# ON-POLICY POLICY GRADIENT REINFORCEMENT LEARNING WITHOUT ON-POLICY SAMPLING

## ABSTRACT

On-policy reinforcement learning (RL) algorithms perform policy updates using i.i.d. trajectories collected by the current policy. However, after observing only a finite number of trajectories, on-policy sampling may produce data that fails to match the expected on-policy data distribution. This *sampling error* leads to noisy updates and data inefficient on-policy learning. Recent work in the policy evaluation setting has shown that non-i.i.d., off-policy sampling can produce data with lower sampling error than on-policy sampling can produce (Zhong et al., 2022). Motivated by this observation, we introduce an adaptive, off-policy sampling method to improve the data efficiency of on-policy policy gradient algorithms. Our method, **P**roximal **R**obust **O**n-**P**olicy **S**ampling (PROPS), reduces sampling error by collecting data with a *behavior policy* that increases the probability of sampling actions that are under-sampled with respect to the current policy. Rather than discarding data from old policies – as is commonly done in on-policy algorithms – PROPS uses data collection to adjust the distribution of previously collected data to be approximately on-policy. We empirically evaluate PROPS on both continuous-action MuJoCo benchmark tasks as well discrete-action tasks and demonstrate that (1) PROPS decreases sampling error throughout training and (2) improves the data efficiency of on-policy policy gradient algorithms. Our work improves the RL community's understanding of a nuance in the on-policy vs off-policy dichotomy: *on-policy learning requires on-policy data, not on-policy sampling.*

## 1 INTRODUCTION

One of the most widely used classes of on-policy methods is the class of on-policy policy gradient algorithms. These algorithms use gradient ascent on the parameters of a parameterized policy so as to increase the probability of observed actions with high expected returns under the current policy. The gradients are commonly estimated using the Monte Carlo estimator, an average computed over i.i.d. samples of trajectories from the current policy. The Monte Carlo estimator is consistent and unbiased; as the number of sampled trajectories increases, the empirical distribution of trajectories converges to the true distribution under the current policy, and thus the empirical gradient converges to the true gradient. However, the expense of environment interactions forces us to work with finite samples. Thus, the empirical distribution of the trajectories often differs from the on-policy sampling distribution. We refer to the mismatch between the empirical distribution of trajectories and the true on-policy trajectory distribution as *sampling error*. This sampling error produces inaccurate gradient estimates, resulting in noisy policy updates and slower learning.

With i.i.d. on-policy sampling, the only way to reduce sampling error is to collect more data. However, on-policy sampling is not explicitly required for on-policy learning; on-policy learning requires on-policy *data* – data whose state-conditioned empirical distribution of actions matches that of the current policy. On-policy sampling is a straightforward way to acquire on-policy data, though we can obtain such data more efficiently *without* on-policy sampling. To better illustrate this concept, consider an MDP with two discrete actions A and B, and suppose the current policy $\pi$ places equal probability on both actions in some state $s$. When following $\pi$, after 10 visits to $s$, we may observe A 2 times and B 8 times rather than the expected 5 times. Alternatively, if we adaptively select the most under-sampled action upon every visit to $s$, we will observe each action an equal number of times. The first scenario illustrates on-policy sampling but not on-policy data; the second scenario uses *off-policy* sampling yet produces on-policy data.

Figure 1: **(Left)** Under on-policy sampling, an agent samples i.i.d. trajectories from its current policy, *i.e.* the data collection policy is simply the current policy. After a policy update, the agent discards these trajectories, since they are now off-policy with respect to the agent's new policy. **(Right)** With PROPS, we retain historic data and adapt the data collection policy so that, when the agent collects new data, the combined dataset matches the expected on-policy distribution.

These observations raise the following question: can on-policy policy gradient algorithms learn more efficiently using on-policy data acquired *without* on-policy sampling? Recently, Zhong et al. (2022) showed that adaptive, off-policy sampling can yield data that more closely matches the on-policy distribution than data produced by i.i.d. on-policy sampling. However, this work was limited to the policy evaluation setting in which the on-policy distribution remains fixed. Turning from evaluation to control poses the challenge of a continually changing current policy.

In this work, we address this challenge and show for the first time that on-policy policy gradient algorithms are more data-efficient learners when they use on-policy data instead of on-policy sampling. Our method, **P**roximal **R**obust **O**n-**P**olicy **S**ampling (PROPS)[1], adaptively corrects sampling error in previously collected data by increasing the probability of sampling actions that are under-sampled with respect to the current policy. As an added bonus, our approach retains historic data from recent older policies for further data efficiency gains. Rather than discarding off-policy data from old policies – as is commonly done in on-policy algorithms – PROPS uses additional data collection to add to the historic data so that the distribution of all data is approximately on-policy. Doing so allows PROPS to leverage off-policy data without requiring off-policy corrections. We provide an overview of our method in Fig. 1. We empirically evaluate PROPS on continuous-action MuJoCo benchmark tasks as well as discrete action tasks and show that (1) PROPS reduces sampling error throughout training and (2) improves the data efficiency of on-policy policy gradient algorithms. In summary, our contributions are

1. We introduce a practical adaptive-sampling-based algorithm that reduces sampling error and permits the reuse of historic data in on-policy algorithms.
2. We demonstrate empirically that our method improves the data efficiency of on-policy policy gradient algorithms.
3. Building off of the theoretical foundation laid by Zhong et al. (2022), this work improves the RL community's understanding of a nuance in the on-policy vs off-policy dichotomy: on-policy learning requires on-policy data, not on-policy sampling.

## 2 RELATED WORK

Our work focuses on data collection in RL. In RL, data collection is often framed as an exploration problem, focusing on how an agent should explore its environment to efficiently learn an optimal policy. Prior RL works have proposed several exploration-promoting methods such as intrinsic motivation (Pathak et al., 2017; Sukhbaatar et al., 2018), count-based exploration (Tang et al., 2017; Ostrovski et al., 2017), and Thompson sampling (Osband et al., 2013; Sutton and Barto, 2018). In contrast, our objective is to learn from the on-policy data distribution; we use adaptive data collection to more efficiently obtain this data distribution.

Prior works have used adaptive off-policy sampling to reduce sampling error in the policy evaluation subfield of RL. Most closely related is the work of Zhong et al. (2022) who first proposed that

---

[1] We include our codebase in the supplemental material.

adaptive off-policy sampling could produce data that more closely matches the on-policy distribution than on-policy sampling could produce. Mukherjee et al. (2022) use a deterministic sampling rule to take actions in a particular proportion. Other bandit works use a non-adapative exploration policy to collect additional data conditioned on previously collected data (Tucker and Joachims, 2022; Wan et al., 2022; Konyushova et al., 2021). As these works only focus on policy evaluation they do not have to contend with a changing on-policy distribution as our work does for the control setting.

Several prior works propose importance sampling methods (Precup, 2000) to reduce sampling error without further data collection. In the RL setting, Hanna et al. (2021) showed that reweighting off-policy data according to an estimated behavior policy can correct sampling error and improve policy evaluation. Similar methods have been studied for temporal difference learning (Pavse et al., 2020) and policy evaluation in the bandit setting (Li et al., 2015; Narita et al., 2019). Conservative Data Sharing (CDS) (Yu et al., 2021) reduces sampling error by selectively integrating offline data from multiple tasks. Our work instead focuses on additional data collection to reduce sampling error.

The method we introduce permits data collected in one iteration of policy optimization to be re-used in future iterations rather than discarded as typically done by on-policy algorithms. Prior work has attempted to avoid discarding data by combining off-policy and on-policy updates with separate loss functions or by using alternative gradient estimates (Wang et al., 2016; Gu et al., 2016; 2017; Fakoor et al., 2020; O'Donoghue et al., 2016; Queeney et al., 2021). In contrast, our method modifies the sampling distribution at each iteration so that the entire data set of past and newly collected data matches the expected distribution under the current policy.

## 3 PRELIMINARIES

### 3.1 REINFORCEMENT LEARNING

We formalize the RL environment as a finite horizon Markov decision process (MDP) (Puterman, 2014) $(\mathcal{S}, \mathcal{A}, p, r, d_0, \gamma)$ with state space $\mathcal{S}$, action space $\mathcal{A}$, transition dynamics $p : \mathcal{S} \times \mathcal{A} \times \mathcal{S} \rightarrow [0, 1]$, reward function $r : \mathcal{S} \times \mathcal{A} \rightarrow \mathbb{R}$, initial state distribution $d_0$, and reward discount factor $\gamma \in [0, 1)$. The state and action spaces may be discrete or continuous. We write $p(\cdot \mid \boldsymbol{s}, \boldsymbol{a})$ to denote the distribution of next states after taking action $\boldsymbol{a}$ in state $\boldsymbol{s}$. We consider stochastic policies $\pi_{\boldsymbol{\theta}} : \mathcal{S} \times \mathcal{A} \rightarrow [0, 1]$ parameterized by $\boldsymbol{\theta}$, and we write $\pi_{\boldsymbol{\theta}}(\boldsymbol{a}|\boldsymbol{s})$ to denote the probability of sampling action $\boldsymbol{a}$ in state $\boldsymbol{s}$ and $\pi_{\boldsymbol{\theta}}(\cdot|\boldsymbol{s})$ to denote the probability distribution over actions in state $\boldsymbol{s}$. The RL objective is to find a policy that maximizes the expected sum of discounted rewards, defined as:

$$J(\boldsymbol{\theta}) = \mathbb{E}_{\boldsymbol{s}_0 \sim d_0, \boldsymbol{a}_t \sim \pi_{\boldsymbol{\theta}}(\cdot|\boldsymbol{s}_t), \boldsymbol{s}_{t+1} \sim p(\cdot|\boldsymbol{s}_t, \boldsymbol{a}_t)} \left[ \sum_{t=0}^{H} \gamma^t r(\boldsymbol{s}_t, \boldsymbol{a}_t) \right], \tag{1}$$

where $H$ is the random variable representing the time-step when an episode ends. Throughout this paper, we refer to the policy used for data collection as the *behavior policy* and the policy trained to maximize its expected return as the *target policy*.

### 3.2 ON-POLICY POLICY GRADIENT ALGORITHMS

Policy gradient algorithms are one of the most widely used methods in RL. These methods perform gradient ascent over policy parameters to maximize an agent's expected return $J(\boldsymbol{\theta})$ (Eq. 1). The gradient of the $J(\boldsymbol{\theta})$ with respect to $\boldsymbol{\theta}$, or *policy gradient*, is often given as:

$$\nabla_{\boldsymbol{\theta}} J(\boldsymbol{\theta}) = \mathbb{E}_{\boldsymbol{s}_0 \sim d_{\pi_{\boldsymbol{\theta}}}, \boldsymbol{a} \sim \pi_{\boldsymbol{\theta}}(\cdot|\boldsymbol{s})} \left[ A^{\pi_{\boldsymbol{\theta}}}(\boldsymbol{s}, \boldsymbol{a}) \nabla_{\boldsymbol{\theta}} \log \pi_{\boldsymbol{\theta}}(\boldsymbol{a}|\boldsymbol{s}) \right], \tag{2}$$

where $A^{\pi_{\boldsymbol{\theta}}}(\boldsymbol{s}, \boldsymbol{a})$ is the *advantage* of choosing action $\boldsymbol{a}$ in state $\boldsymbol{s}$ and following $\pi_{\boldsymbol{\theta}}$ thereafter. In practice, the expectation in Eq. 2 is approximated with Monte Carlo samples collected from $\pi_{\boldsymbol{\theta}}$ and an estimate of $A^{\pi_{\boldsymbol{\theta}}}$ used in place of the true advantages (Schulman et al., 2016). After updating the policy parameters with this estimated gradient, the previously collected trajectories $\mathcal{D}$ become off-policy with respect to the updated policy. To ensure gradient estimation remains unbiased, on-policy algorithms discard historic data after each update and collect new trajectories with the updated policy.

If we look at the fundamentals of policy evaluation in dynamic programming we see a very clear picture of the ideal update (i.e. using the probabilities of the state distribution and policies directly), but at the cost of knowing the transition probabilities and having to sweep over all states and actions

for a single update. When improving the policy we go between a policy improvement step (i.e. maximize the learned value function) and the policy evaluation step. Generalized policy iteration made this update more general by enabling the ability to improve a policy without running policy evaluation for every state and action. From generalized policy iteration we can get to many of our on-policy algorithms such as TD, sarsa, on-policy actor-critic algorithms, and many others with the inclusion of sampling from distributions instead of knowing the distributions ahead of time. This type of on-policy learning uses a transition and throws it away, meaning any policy improvement done will always appear in the data and there is no stale buffer of data. From this trajectory of literature it is clear, on-policy algorithms are designed and work well with data that is distributed according to the target policy. The easiest way to get this data is through sampling according to the behavior.

This foundational idea of policy learning via stochastic gradient ascent was first proposed by Williams (Williams, 1992) under the name REINFORCE. Since then, a large body of research has focused on developing more scalable policy gradient methods (Kakade, 2001; Schulman et al., 2015; Mnih et al., 2016; Espeholt et al., 2018; Lillicrap et al., 2015; Haarnoja et al., 2018). Arguably, the most successful variant of policy gradient learning is proximal policy optimization (PPO) (Schulman et al., 2017), the algorithm of choice in several high-profile success stories (Berner et al., 2019; Akkaya et al., 2019; Vinyals et al., 2019). Rather than maximizing the standard RL objective (Eq. 1), PPO maximizes a surrogate objective:

$$\mathcal{L}_{\text{PPO}}(\boldsymbol{s}, \boldsymbol{a}, \boldsymbol{\theta}, \boldsymbol{\theta}_{\text{old}}) = \min(g(\boldsymbol{s}, \boldsymbol{a}, \boldsymbol{\theta}, \boldsymbol{\theta}_{\text{old}})A^{\pi_{\boldsymbol{\theta}_{\text{old}}}}(\boldsymbol{s}, \boldsymbol{a}),$$
$$\text{clip}(g(\boldsymbol{s}, \boldsymbol{a}, \boldsymbol{\theta}, \boldsymbol{\theta}_{\text{old}}), 1 - \epsilon, 1 + \epsilon)A^{\pi_{\boldsymbol{\theta}_{\text{old}}}}(\boldsymbol{s}, \boldsymbol{a})), \tag{3}$$

where $\boldsymbol{\theta}_{\text{old}}$ denotes the policy parameters prior to the update, $g(\boldsymbol{s}, \boldsymbol{a}, \boldsymbol{\theta}, \boldsymbol{\theta}_{\text{old}})$ is the policy ratio $g(\boldsymbol{s}, \boldsymbol{a}, \boldsymbol{\theta}, \boldsymbol{\theta}_{\text{old}}) = \frac{\pi_{\boldsymbol{\theta}}(\boldsymbol{a}|\boldsymbol{s})}{\pi_{\boldsymbol{\theta}_{\text{old}}}(\boldsymbol{a}|\boldsymbol{s})}$, and the clip function with hyperparameter $\epsilon$ clips $g(\boldsymbol{s}, \boldsymbol{a}, \boldsymbol{\theta}, \boldsymbol{\theta}_{\text{old}})$ to the interval $[1 - \epsilon, 1 + \epsilon]$. The first term inside the minimum of $\mathcal{L}_{\text{PPO}}$ is the conservative policy iteration (CPI) objective (Kakade and Langford, 2002). The second term clips the CPI objective to disincentivizes large changes to $\pi_{\boldsymbol{\theta}}(\boldsymbol{a}|\boldsymbol{s})$. In contrast to other policy gradient algorithms which perform a single gradient update per data sample to avoid destructively large weight updates, PPO's clipping mechanism allows the agent to perform multiple epochs of minibatch policy updates.

## 4 CORRECTING SAMPLING ERROR IN REINFORCEMENT LEARNING

In this section, we describe two potential sources of sampling error that arise when computing policy gradient estimates and then describe how adaptive, off-policy sampling can reduce such sampling error. The first source arises from the fact that some actions will almost inevitably be under-sampled during on-policy sampling and thus contribute less to the gradient estimate than they should in expectation. The second source arises when collected data is *not* discarded after each policy update. While most on-policy algorithms bypass this second source by simply discarding data, we include its mention here as we will show that adjusting for this source of sampling error with additional data collection – instead of discarding data – further increases data efficiency.

As a motivating example, consider training an agent to walk using an on-policy policy gradient algorithm. For exposition's sake, we assume that trajectories can be categorized into three types: successful walking, standing in place, and falling over. We show a portion of the task's MDP in Fig. 2, including the agent's probability of experiencing each outcome following its initial policy as well as each outcome's return. From an initial standing state, the agent chooses either to attempt to move or to remain standing in place. If the agent moves, it has a large probability of falling and a small probability of successfully walking; if it stays stationary then it will always end with the standing outcome. [2]

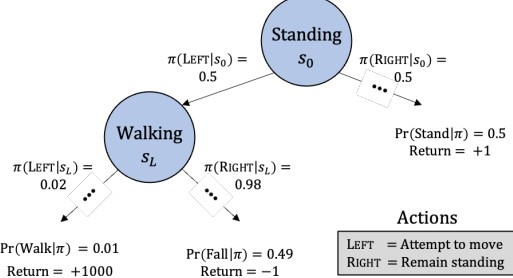

Figure 2: A portion of an MDP for a walking task with three possible outcomes: walking, standing, and falling.

---

[2]This example is intentionally simplified for the purpose of illustration. In reality, the agent could experience many different outcomes and the trajectories we describe would contain possibly hundreds of smaller actions.

Walking yields a much larger return than standing or falling but is unlikely under the current policy because the probability of attempting to move at $s_L$ is low; after sampling 100 trajectories, this critical action may be entirely absent from the observed data. Having never attempted to move at $s_L$, the agent will believe that moving at $s_0$ always leads to a return of $-1$ while not moving leads to a return of $+1$. Thus, a Monte Carlo estimate of the policy gradient reduces the probability of attempting to move at $s_0$, making the agent even less likely to observe walking and more likely to converge to a suboptimal standing policy.

For the agent in our example to learn to walk, all actions must be observed closer to their true probabilities under the current policy. The crux of this issue is that randomness in the i.i.d sampling of on-policy trajectories leads to a mismatch in the empirical distribution over actions and the true on-policy distribution. As such, low probability actions that lead to high return can be under-sampled. We refer to this mismatch as *sampling error*. Sampling error is an inherent feature of i.i.d. on-policy sampling; only in the limit of infinite data will such sampling achieve zero sampling error, but the expense of environment interaction forces us to use finite samples during RL training. In principle, we can increase the data available at each iteration of policy improvement by reusing data collected during earlier iterations. However, such data is off-policy and its inclusion would generally increase sampling error and bias gradient estimates. One can introduce off-policy corrections to reduce gradient bias (Wang et al., 2016; Gu et al., 2016; 2017; Queeney et al., 2021), but such approaches may increase the variance of gradient estimates.

Adaptive sampling provides an alternative means of producing on-policy data without an infinite sample size. In terms of our example, suppose the agent in Fig. 2 observes 49 falling and 50 standing trajectories such that the "attempt to move" action is under-sampled at $s_0$ and $s_L$ with respect to the true distribution of actions under the agent's policy. If the agent samples an additional trajectory from the on-policy distribution, it has a low probability of sampling this action at both $s_0$ and $s_L$ and thus a low probability of observing walking. However, if instead the agent increases the probability of attempting to move in states $s_0$ and $s_L$, then it is more likely to observe these under-sampled actions and hence have low sampling error in the entire dataset of 100 trajectories. While the adaptive method uses off-policy sampling, it produces data that is more on-policy than the data produced from on-policy sampling. In the following section, we leverage this concept to develop a new adaptive sampling algorithm that reduces sampling error in on-policy policy gradient algorithms.

# 5 PROXIMAL ROBUST ON-POLICY SAMPLING FOR POLICY GRADIENT ALGORITHMS

Our goal is to develop an adaptive, off-policy sampling algorithm that reduces sampling error in on-policy policy gradient estimates. We outline a general framework for on-policy learning with an adaptive behavior policy in Algorithm 1. In this framework, the behavior policy $\pi_\phi$ and target policy $\pi_\theta$ are initially the same. The behavior policy collects a batch of $m$ transitions, adds the batch to a replay buffer $\mathcal{D}$, and then updates its weights such that the next batch it collects reduces sampling error in $\mathcal{D}$ with respect to the target policy $\pi_\theta$ (Lines 7-10). Every $n$ steps (with $n > m$), the agent updates its target policy with data from $\mathcal{D}$ (Line 11). We refer to $m$ and $n$ as

---

**Algorithm 1** On-policy policy gradient algorithm with adaptive sampling

1: **Inputs**: Target batch size $n$, behavior batch size $m$, buffer size $b$.
2: **Output:** Target policy parameters $\theta$.
3: Initialize target policy parameters $\theta$.
4: Initialize behavior policy parameters $\phi \leftarrow \theta$.
5: Initialize empty replay buffer $\mathcal{D}$ with capacity $bn$.
6: **for** target update $i = 1, 2, \ldots$ **do**
7:     **for** behavior update $j = 1, \ldots, \lfloor n/m \rfloor$ **do**
8:         Collect batch of data $\mathcal{B}$ by running $\pi_\phi$.
9:         Append $\mathcal{B}$ to replay buffer $\mathcal{D}$.
10:        Update $\pi_\phi$ with $\mathcal{D}$ using Algorithm 2.
11:    Update $\pi_\theta$ with $\mathcal{D}$.
12: **return** $\theta$

---

the *behavior batch size* and the *target batch size*, respectively. The replay buffer can hold up to $b$ target batches ($bn$ transitions), and we call $b$ the *buffer size*. Regardless of how $b$ is set, the role of the behavior policy is to continually adjust action probabilities for new samples so that the aggregate data distribution of $\mathcal{D}$ mathces the expected on-policy distribution of the current target policy (Line 10). Implementing Line 10 is the core challenge we address in the remainder of this section.

To ensure that the empirical distribution of $\mathcal{D}$ matches the expected on-policy distribution, updates to $\pi_\phi$ should attempt to increase the probability of actions which are currently under-sampled with respect to $\pi_\theta$. Zhong et al. (2022) recently developed a simple method called Robust On-policy Sampling (ROS) for making such updates. In particular, the gradient $\nabla_\phi \mathcal{L} := -\nabla_\phi \sum_{(s,a) \in \mathcal{D}} \log \pi_\phi(a|s)$ *when evaluated at $\phi = \theta$* provides a direction to change $\phi$ such that under-sampled actions have their probabilities increased. Thus a single step of gradient ascent will increase the probability of under-sampled actions.[3] In theory and in simple RL policy evaluation tasks, this update was shown to improve the rate at which the empirical data distribution converges to the on-policy distribution. Unfortunately, there are two main challenges that render ROS unsuitable for Line 10 in Algorithm 1.

**Destructively large policy updates.** Since the replay buffer $\mathcal{D}$ may contain data collected from older target policies, some samples in $\mathcal{D}$ may be very off-policy with respect to the current target policy such that $\log \pi_\phi(a|s)$ is large and negative. Since $\nabla_\phi \log \pi_\phi(a|s)$ increases in magnitude as $\pi_\phi(a|s)$ tends towards zero, ROS incentivizes the agent to continually decrease the probability of these actions despite being extremely unlikely under the current target policy. Thus, off-policy samples can produce destructively large policy updates.

**Improper handling of continuous actions.** In a continuous-action task, ROS may produce behavior policies that *increase* sampling error. A continuous-action task policy $\pi_\theta(a|s)$ is typically parameterized as a Gaussian $\mathcal{N}(\mu(s), \Sigma(s))$ with mean $\mu(s)$ and diagonal covariance matrix $\Sigma(s)$. Since actions in the tail of the Gaussian far from the mean will always be under-sampled, the ROS update will continually push the components of $\mu(s)$ towards $\pm\infty$ and the diagonal components of $\Sigma(s)$ towards 0 to increase the probability of sampling these actions. The result is a degenerate behavior policy that is so far from the target policy that sampling from it increases sampling error. We illustrate this scenario with 1-dimensional continuous actions in Fig. 6 of Appendix A.

To resolve these issues, we propose a new behavior policy update. Since the gradient of the ROS loss $\nabla_\phi \mathcal{L} = \nabla_\phi \log \pi_\phi(a|s)|_{\phi=\theta}$ resembles the policy gradient (Eq. 2) with $A^{\pi_\theta}(s, a) = -1, \forall(s, a)$, we draw inspiration from PPO (Schulman et al., 2017) to address these challenges. Rather than using the ROS objective to update $\phi$, we use a clipped surrogate objective:

$$\mathcal{L}_{\text{CLIP}}(s, a, \phi, \theta, \epsilon_{\text{PROPS}}) = \min\left[-\frac{\pi_\phi(a|s)}{\pi_\theta(a|s)}, -\texttt{clip}\left(\frac{\pi_\phi(a|s)}{\pi_\theta(a|s)}, 1 - \epsilon_{\text{PROPS}}, 1 + \epsilon_{\text{PROPS}}\right)\right]. \quad (4)$$

Intuitively, this objective incentivizes the agent to decrease the probability of observed actions by at most a factor of $1 - \epsilon_{\text{PROPS}}$. Let $g(s, a, \phi, \theta) = \frac{\pi_\phi(a|s)}{\pi_\theta(a|s)}$. When $g(s, a, \phi, \theta) < 1 - \epsilon_{\text{PROPS}}$, this objective is clipped at $-(1 - \epsilon_{\text{PROPS}})$. The loss gradient $\nabla_\phi \mathcal{L}_{\text{CLIP}}$ becomes zero, and the $(s, a)$ pair has no effect on the policy update. When $g(s, a, \phi, \theta) > 1 - \epsilon_{\text{PROPS}}$, clipping does not apply, and the (non-zero) gradient $\nabla_\phi \mathcal{L}_{\text{CLIP}}$ points in a direction that decreases the probability of $\pi_\phi(a|s)$. As in the PPO update, this clipping mechanism avoids destructively large policy updates and permits us to perform many epochs of minibatch updates with the same batch of data.

To address the second challenge and prevent degenerate behavior policies, we introduce an auxiliary loss that incentivizes the agent to minimize the KL divergence between the behavior policy and target policy at states in the observed data. The full PROPS objective is then:

$$\mathcal{L}_{\text{PROPS}}(s, a, \phi, \theta, \epsilon_{\text{PROPS}}, \lambda) = \mathcal{L}_{\text{CLIP}}(s, a, \phi, \theta) - \lambda D_{\text{KL}}(\pi_\theta(\cdot|s)||\pi_\phi(\cdot|s)) \quad (5)$$

where $\lambda$ is a regularization coefficient quantifying a trade-off between maximizing $\mathcal{L}_{\text{PROPS}}$ and minimizing $D_{\text{KL}}$. We provide full pseudocode for the PROPS update in Algorithm 2. Like ROS, we set the behavior policy parameters $\phi$ equal to the target policy parameters, and then make a local adjustment to $\phi$ to increase the probabilities of under-sampled actions. We stop the PROPS update when $D_{\text{KL}}(\pi_\theta||\pi_\phi)$ reaches a chosen threshold $\delta_{\text{PROPS}}$. This technique further safeguards against large policy updates and is used in widely adopted implementations of PPO (Raffin et al., 2021; Liang et al., 2018). The PROPS update allows us to efficiently learn a behavior policy that keeps the distribution of data in the replay buffer close to the expected distribution of the target policy.

## 6 EXPERIMENTS

---

[3]To add further intuition for this update, note that it is the opposite of a gradient ascent step on the log likelihood of $\mathcal{D}$. When starting at $\theta$, gradient ascent on the data log likelihood will increase the probability of actions that are over-sampled relative to $\pi_\theta$. Hence, the ROS update changes $\phi$ in the opposite direction.

The central goal of our work is to understand whether on-policy data or on-policy sampling is critical to on-policy policy gradient learning. Towards this goal, we design experiments to answer the two questions: **(Q1)** Does PROPS achieve lower sampling error than on-policy sampling during policy gradient learning (*i.e.*, does PROPS produces data that is more on-policy than on-policy sampling produces)? **(Q2)** Does PROPS improve the data efficiency of on-policy algorithms? Our empirical analysis focuses on continuous-action MuJoCo benchmark tasks and three discrete-action tasks: CartPole-v1, LunarLander-v2, and Discrete2D100-v0 – a 2D navigation task with 100 discrete actions. We include results for discrete action tasks in Appendix E.

---

**Algorithm 2** PROPS Update

1: **Inputs:** Target policy parameters $\boldsymbol{\theta}$, replay buffer $\mathcal{D}$, target KL $\delta$, clipping coefficient $\epsilon_{\text{PROPS}}$, regularizer coefficient $\lambda$, n_epoch, n_minibatch.
2: **Output:** Behavior policy parameters $\phi$.
3: $\phi \leftarrow \boldsymbol{\theta}$
4: **for** epoch $i = 1, 2, \ldots,$ n_epoch **do**
5:      **for** minibatch $j = 1, 2, \ldots,$ n_minibatch **do**
6:          Sample minibatch $\mathcal{D}_j \sim \mathcal{D}$
7:          Compute the loss (Eq. 5)

$$\mathcal{L} \leftarrow \frac{1}{|\mathcal{D}_j|} \sum_{(\boldsymbol{s}, \boldsymbol{a}) \in \mathcal{D}_j} \mathcal{L}_{\text{PROPS}}(\boldsymbol{s}, \boldsymbol{a}, \phi, \boldsymbol{\theta}, \epsilon_{\text{PROPS}}, \lambda)$$

8:          Update $\phi$ with a step of gradient ascent on $\mathcal{L}$
9:          **if** $D_{\text{KL}}(\pi_{\boldsymbol{\theta}} || \pi_{\phi}) > \delta_{\text{PROPS}}$ **then**
10:             **return** $\phi$
11: **return** $\phi$

---

## 6.1 CORRECTING SAMPLING ERROR FOR A FIXED TARGET POLICY

We first study how quickly PROPS decreases sampling error in MuJoCo benchmark tasks when the target policy is fixed. This setting is similar to the policy evaluation setting considered by Zhong et al. (2022). As such, we provide two baselines for comparison: on-policy sampling and ROS. We follow Zhong et al. (2022) and measure sampling error using the KL-divergence $D_{\text{KL}}(\pi_{\mathcal{D}} || \pi_{\boldsymbol{\theta}})$ between the empirical policy $\pi_{\mathcal{D}}$ and the target policy $\pi_{\boldsymbol{\theta}}$. We estimate $\pi_{\mathcal{D}}$ as the maximum likelihood estimate under data in the replay buffer via stochastic gradient ascent. Further details on how we compute $\pi_{\mathcal{D}}$ are in Appendix B.

In all tasks, we set the behavior batch size to $m = 256$, use a replay buffer with capacity of $64m$ samples, and collect a total of $128m$ samples. Thus, we expect sampling error to decrease over the first $64m$ samples and then remain roughly constant afterwards. We tune ROS and PROPS separately for each task using a hyperparameter sweep described in Appendix C and report results for hyperparameters yielding the lowest sampling error.

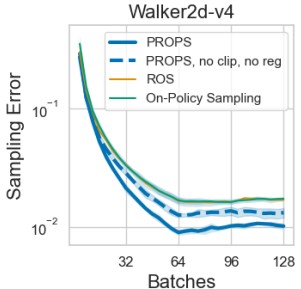

Figure 3: PROPS reduces sampling error faster than on-policy sampling and ROS. The ROS and on-policy sampling curves overlap. Solid curves denote means over 5 seeds. Shaded regions denote 95% confidence intervals.

As shown in Fig. 3, PROPS decreases sampling error faster than on-policy sampling and ROS. In fact, ROS shows little to no improvement over on-policy sampling in every task. This limitation of ROS is unsurprising, as Zhong et al. (2022) showed that ROS struggled to reduce sampling error even in low-dimensional continuous-action tasks. Moreover, PROPS decreases sampling error without clipping and regularization, emphasizing how adaptive off-policy sampling alone decreases sampling error. Due to space constraints, we include results for the remaining environments in Appendix C. We additionally include experiments using a fixed, randomly initialized target policy as well as ablation studies isolating the effects of PROPS's objective clipping and regularization in Appendix C. Results with a random target policy are qualitatively similar to those in Fig. 3, and we observe that clipping and regularization both individually help reduce sampling error.

## 6.2 CORRECTING SAMPLING ERROR DURING RL TRAINING

We are ultimately interested in understanding how replacing on-policy sampling with PROPS affects the data efficiency of on-policy learning, where the target policy is continually updated. In the following experiments, we train RL agents with PROPS and on-policy sampling to evaluate (1) the data efficiency of training and (2) the sampling error $D_{\text{KL}}(\pi_{\mathcal{D}} || \pi_{\boldsymbol{\theta}})$ throughout training. We measure

data efficiency as the return achieved within a fixed training budget. Since ROS (Zhong et al., 2022) is expensive and fails to reduce sampling error even with a fixed policy, we omit it from this analysis.

We use PPO (Schulman et al., 2017) to update the target policy. We consider two baseline methods for providing data to compute PPO updates: (1) vanilla PPO with on-policy sampling, and (2) PPO with on-policy sampling and a replay buffer of size $b$ (PPO-BUFFER). PPO-BUFFER is a naive method for improving data efficiency of on-policy algorithms by reusing off-policy data collected by old target policies as if it were on-policy data. Although PPO-BUFFER will have biased gradients, it has been successfully applied in difficult learning tasks (Berner et al., 2019).

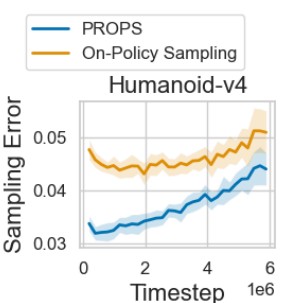

In these experiments, we use a buffer size of $b = 2$ such that agents retain a batch of data for one extra iteration before discarding it. Since PROPS and PPO-BUFFER compute target policy updates with $b$ times as much learning data as PPO, we integrate this extra data by increasing the minibatch size for target and behavior policy updates by a factor of $b$. To ensure a fair comparison between algorithms, we tune each algorithm separately using a hyperparameter sweep described in Appendix F. We report results for the hyperparameters yielding the largest final return. For each algorithm and task, we plot the interquartile mean (IQM) return throughout training as well as the distribution of returns achieved at the end of training (*i.e.*, the performance profile) (Agarwal et al., 2021).

Figure 4: PROPS achieves lower sampling error than on-policy sampling during RL training. Curves denote averages over 50 seeds. Shaded regions denote 95% confidence intervals.

As shown in Fig. 5a, PROPS achieves higher return than both PPO and PPO-BUFFER throughout training in all tasks except Ant-v4 where PROPS dips slightly below PPO's return near the end of training. Moreover, in Fig. 5b, the performance profile of PROPS almost always lies above the performance profiles of PPO and PPO-BUFFER. Thus, we affirmatively answer **Q2** posed at the start of this section: PROPS increases data efficiency.

In Appendix E, we provide additional experiments demonstrating that PROPS improves data efficiency in discrete-action tasks. We additionally ablate the buffer size $b$ in Appendix D. We find that data efficiency may decrease with a larger buffer size. Intuitively, the more historic data kept around, the more data that must be collected to impact the aggregate data distribution.

Having established that PROPS improves data efficiency, we now investigate if PROPS is appropriately adjusting the data distribution of the replay buffer by comparing the sampling error achieved throughout training with PROPS and PPO-BUFFER. Training with PROPS produces a different sequence of target policies than training with PPO-BUFFER produces. To provide a fair comparison, we compute sampling error for PPO-BUFFER using the target policy sequence produced by PROPS. More concretely, we fill a second replay buffer with on-policy samples collected by the *target policies* produced while training with PROPS and then compute the sampling error using data in this buffer.

As shown in Fig. 4, PROPS achieves lower sampling error than on-policy sampling with a replay buffer. Due to space constraints, we provide sampling error curves for the remaining environments in Appendix D. We ablate the effects of the clipping coefficient $\epsilon_{\text{PROPS}}$ and regularization coefficient $\lambda$ in Appendix D. Without clipping or without regularization, PROPS often achieves greater sampling error than on-policy sampling, indicating that both help to keep sampling error low. Moreover, data efficiency generally decreases when we remove clipping or regularization, showing both are essential to PROPS. Thus, we affirmatively answer **Q1** posed at the start of this section: PROPS achieves lower sampling error than on-policy sampling when the target policy is fixed and during RL training.

## 7   DISCUSSION

This work has shown that adaptive, off-policy sampling can be used to reduce sampling error in data collected throughout RL training and improve the data efficiency of on-policy policy gradient algorithms. We have introduced an algorithm that scales adaptive off-policy sampling to continuous control RL benchmarks and enables tracking of the changing on-policy distribution. By integrating this data collection procedure into the popular PPO algorithm, the main conclusion of our analysis is

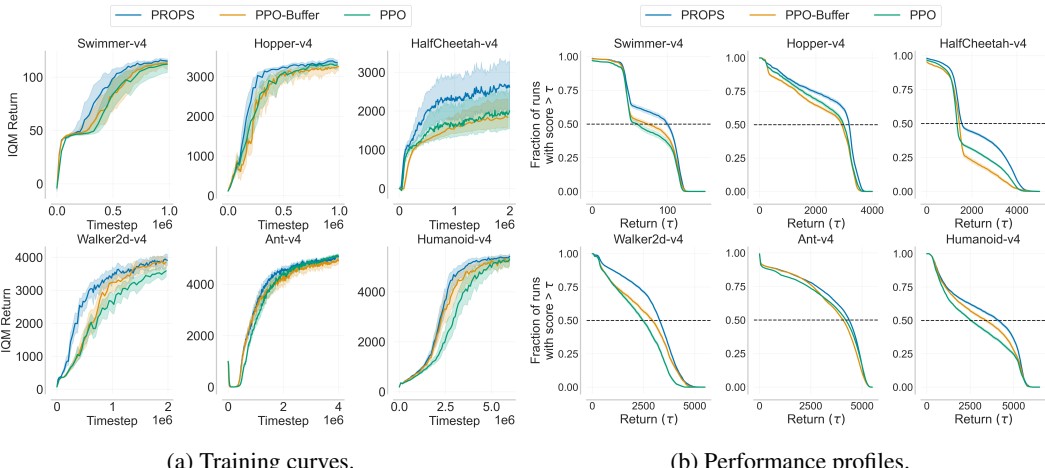

(a) Training curves.                    (b) Performance profiles.

Figure 5: **(a)** IQM returns over 50 seeds. Shaded regions denote 95% bootstrapped confidence intervals. **(b)** Performance profiles over 50 seeds. The return $\tau$ for which the profiles intersect $y = 0.5$ is the median, and the area under the performance profile corresponds to the mean. Shaded regions denote 95% bootstrapped confidence intervals.

that on-policy learning algorithms learn most efficiently with on-policy data, *not* on-policy sampling. In this section, we discuss limitations of our work and present opportunities for future research.

PROPS builds upon the ROS algorithm of Zhong et al. (2022). While Zhong et al. (2022) focused on theoretical analysis and policy evaluation in small scale domains, we chose to focus on empirical analysis with policy learning in standard RL benchmarks. An important direction for future work would be theoretical analysis of PROPS, in particular whether PROPS also enjoys the same faster convergence rate that was shown for ROS relative to on-policy sampling.

A limitation of PROPS is that the update indiscriminately increases the probability of under-sampled actions without considering their importance in gradient computation. For instance, if an under-sampled action has zero advantage, it has no impact on the gradient and need not be sampled. An interesting direction for future work could be to prioritize correcting sampling error for $(s, a)$ that have the largest influence on the gradient estimate, *i.e.*, large advantage (positive or negative).

Beyond these more immediate directions, our work opens up other opportunities for future research. A less obvious feature of the PROPS behavior policy update is that it can be used track the empirical data distribution of *any* desired policy, not only that of the current policy. This feature means PROPS has the potential to be integrated into off-policy RL algorithms and used so that the empirical distribution more closely matches a desired exploration distribution. Thus, PROPS could be used to perform focused exploration without explicitly tracking state and action counts.

# 8 CONCLUSION

In this paper, we ask whether on-policy policy gradient methods are more data efficient using on-policy sampling or on-policy *data* acquired *without* on-policy sampling. To answer this question, we introduce an adaptive, *off-policy* sampling method for on-policy policy gradient learning that collects data such that the empirical distribution of sampled actions closely matches the expected on-policy data distribution at observed states. Our method, Proximal Robust On-policy Sampling (PROPS), periodically updates the data collecting behavior policy so as to increase the probability of sampling actions that are currently under-sampled with respect to the on-policy distribution. Furthermore, rather than discarding collected data after every policy update, PROPS permits more data efficient on-policy learning by using data collection to adjust the distribution of previously collected data to be approximately on-policy. We replace on-policy sampling with PROPS to generate data for the widely-used PPO algorithm and empirically demonstrate that PROPS produces data that more closely matches the expected on-policy distribution and yields more data efficient learning compared to on-policy sampling.

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

| $g(\boldsymbol{s}\,\boldsymbol{a},\boldsymbol{\phi},\boldsymbol{\theta}) > 0$ | Is the objective clipped? | Return value of $\min$ | Gradient |
|---|---|---|---|
| $g(\boldsymbol{s}\,\boldsymbol{a},\boldsymbol{\phi},\boldsymbol{\theta}) \in [1 - \epsilon_{\text{PROPS}}, 1 + \epsilon_{\text{PROPS}}]$ | No | $-g(\boldsymbol{s},\boldsymbol{a},\boldsymbol{\phi},\boldsymbol{\theta})$ | $\nabla_{\boldsymbol{\phi}}\mathcal{L}_{\text{CLIP}}$ |
| $g(\boldsymbol{s},\boldsymbol{a},\boldsymbol{\phi},\boldsymbol{\theta}) > 1 + \epsilon_{\text{PROPS}}$ | No | $-g(\boldsymbol{s},\boldsymbol{a},\boldsymbol{\phi},\boldsymbol{\theta})$ | $\nabla_{\boldsymbol{\phi}}\mathcal{L}_{\text{CLIP}}$ |
| $g(\boldsymbol{s},\boldsymbol{a},\boldsymbol{\phi},\boldsymbol{\theta}) < 1 - \epsilon_{\text{PROPS}}$ | Yes | $-(1 - \epsilon_{\text{PROPS}})$ | $\mathbf{0}$ |

Table 1: Behavior of PROPS's clipped surrogate objective (Eq. 4).

## A  PROPS Implementation Details

In this appendix, we describe two relevant implementation details for the PROPS update (Algorithm 2). We additionally summarize the behavior of PROPS's clipping mechanism in Table 1.

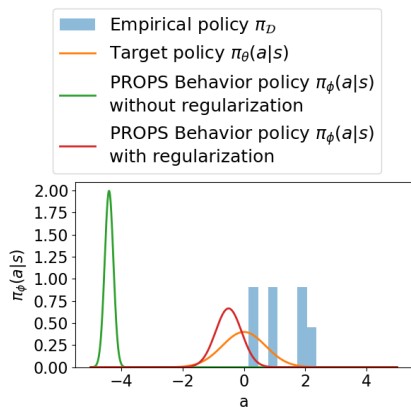

1. **PROPS update:** The PROPS update adapts the behavior policy to reduce sampling error in the replay buffer $\mathcal{D}$. When performing this update with a full replay buffer, we exclude the oldest batch of data collected by the behavior policy (*i.e.*, the $m$ oldest transitions in $\mathcal{D}$); this data will be evicted from the replay buffer before the next behavior policy update and thus does not contribute to sampling error in $\mathcal{D}$.

2. **Behavior policy class:** We compute behavior policies from the same policy class used for target policies. In particular, we consider Gaussian policies which output a mean $\mu(\boldsymbol{s})$ and a variance $\sigma^2(\boldsymbol{s})$ and then sample actions $\boldsymbol{a} \sim \pi(\cdot|\boldsymbol{s}) \equiv \mathcal{N}(\mu(\boldsymbol{s}), \sigma^2(\boldsymbol{s}))$. In principle, the target and behavior policy classes can be different. However, using the same class for both policies allows us to easily initialize the behavior policy equal to the target policy at the start of each update. This initialization is necessary to ensure the PROPS update increases the probability of sampling actions that are currently under-sampled with respect to the target policy.

Figure 6: In this example, $\pi(\cdot|\boldsymbol{s}) = \mathcal{N}(0, 1)$. After several visits to $\boldsymbol{s}$, all sampled actions (blue) satisfy $a > 0$ so that actions $a < 0$ are under-sampled. Without regularization, PROPS will attempt to increase the probabilities of under-sampled action in the tail of target policy distribution (green). The regularization term in the PROPS objective ensures the behavior policy remains close to target policy.

## B  Computing Sampling Error

We claim that PROPS improves the data efficiency of on-policy learning by reducing sampling error in the agent's replay buffer $\mathcal{D}$ with respect to the agent's current (target) policy. To measure sampling error, we use the KL-divergence $D_{\text{KL}}(\pi_{\mathcal{D}}||\pi_{\boldsymbol{\theta}})$ between the empirical policy $\pi_{\mathcal{D}}$ and the target policy $\pi_{\boldsymbol{\theta}}$ which is the primary metric Zhong et al. (2022) used to show ROS reduces sampling error:

$$D_{\text{KL}}(\pi_{\mathcal{D}}||\pi_{\boldsymbol{\theta}}) = \mathbb{E}_{\boldsymbol{s}\sim\mathcal{D},\boldsymbol{a}\sim\pi_{\mathcal{D}}(\cdot|\boldsymbol{s})}\left[\log\left(\frac{\pi_{\mathcal{D}}(\boldsymbol{a}|\boldsymbol{s})}{\pi_{\boldsymbol{\theta}}(\boldsymbol{a}|\boldsymbol{s})}\right)\right]. \tag{6}$$

We compute a parametric estimate of $\pi_{\mathcal{D}}$ by maximizing the log-likelihood of $\mathcal{D}$ over the same policy class used for $\pi_{\boldsymbol{\theta}}$. More concretely, we let $\boldsymbol{\theta}'$ be the parameters of neural network with the same architecture as $\pi_{\boldsymbol{\theta}}$ train and then compute:

$$\boldsymbol{\theta}_{\text{MLE}} = \arg\max_{\boldsymbol{\theta}'} \sum_{(\boldsymbol{s},\boldsymbol{a})\in\mathcal{D}} \log \pi_{\boldsymbol{\theta}'}(\boldsymbol{a}|\boldsymbol{s}) \tag{7}$$

using stochastic gradient ascent. After computing $\boldsymbol{\theta}_{\text{MLE}}$, we then estimate sampling error using the Monte Carlo estimator:

$$D_{\text{KL}}(\pi_{\mathcal{D}}||\pi_{\boldsymbol{\theta}}) \approx \sum_{(\boldsymbol{s},\boldsymbol{a})\in\mathcal{D}} \left(\log \pi_{\boldsymbol{\theta}_{\text{MLE}}}(\boldsymbol{a}|\boldsymbol{s}) - \log \pi_{\boldsymbol{\theta}}(\boldsymbol{a}|\boldsymbol{s})\right). \tag{8}$$

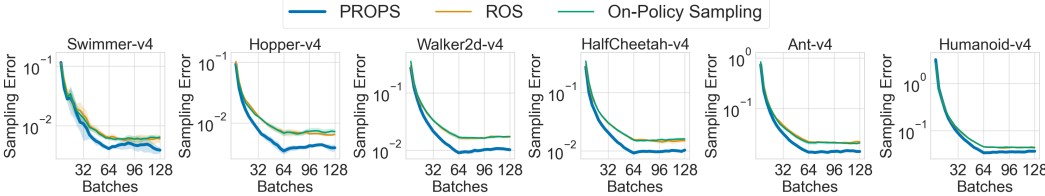

Figure 7: Sampling error with a fixed, expert target policy. Solid curves denote the mean over 5 seeds. Shaded regions denote 95% confidence belts.

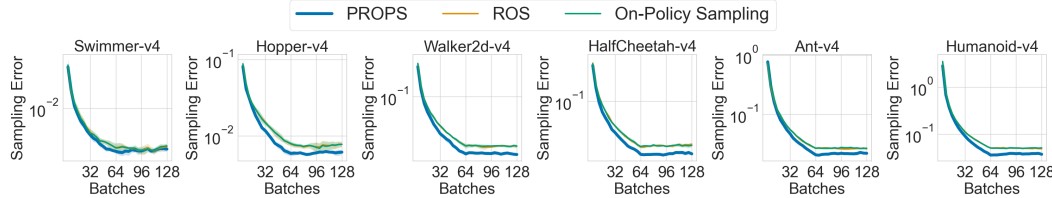

Figure 8: Sampling error with a fixed, randomly initialized target policy. Solid curves denote the mean over 5 seeds. Shaded regions denote 95% confidence belts.

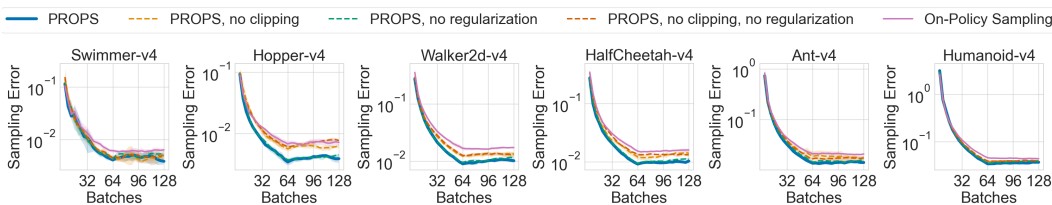

Figure 9: Sampling error ablations with a fixed, expert target policy. Here, "no clipping" refers to setting $\epsilon_{\text{PROPS}} = \infty$, and "no regularization" refers to setting $\lambda = 0$. Solid curves denote the mean over 5 seeds, and shaded regions denote $\pm$ one standard error.

## C  CORRECTING SAMPLING ERROR FOR A FIXED TARGET POLICY

In this appendix, we expand upon results presented in Section 6.1 of the main paper and provide additional experiments investigating the degree to which PROPS reduces sampling error with respect to a fixed target policy. We include empirical results for all six MuJoCo benchmark tasks as well as ablation studies investigating the effects of clipping and regularization.

We tune PROPS and ROS using a hyperparameter sweep. For PROPS, we consider learning rates in $\{10^{-3}, 10^{-4}\}$, regularization coefficients $\lambda \in \{0.01, 0.1, 0.3\}$, and PROPS target KLs in $\delta_{\text{PROPS}} \in \{0.05, 0.1\}$. We fix $\epsilon_{\text{PROPS}} = 0.3$ across all experiments. For ROS, we consider learning rates in $\{10^{-3}, 10^{-4}, 10^{-5}\}$. We report results for the hyperparameters yielding the lowest sampling error.

Fig. 7 and 8 show sampling error computed with a fixed expert and randomly initialized target policy, respectively. We see that PROPS achieves lower sampling error than both ROS and on-policy sampling across all tasks. ROS shows little to no improvement over on-policy sampling, highlighting the difficulty of applying ROS to higher dimensional tasks with continuous actions.

Fig. 9 ablates the effects of PROPS's clipping mechanism and regularization on sampling error reduction. We ablate clipping by setting $\epsilon_{\text{PROPS}} = \infty$, and we ablate regularization by setting $\lambda = 0$. We use a fixed expert target policy and use the same tuning procedure described earlier in this appendix. In all tasks, PROPS achieves higher sampling error without clipping nor regularization than it does with clipping and regularization. However, it nevertheless outperforms on-policy sampling in

all tasks except Hopper where it matches the performance of on-policy sampling. Only including regularization slightly decreases sampling error, whereas clipping alone produces sampling error only slightly higher than that achieved by PROPS with both regularization and clipping. These observations indicate that while regularization in is helpful, clipping has a stronger effect on sampling error reduction than regularization when the target policy is fixed.

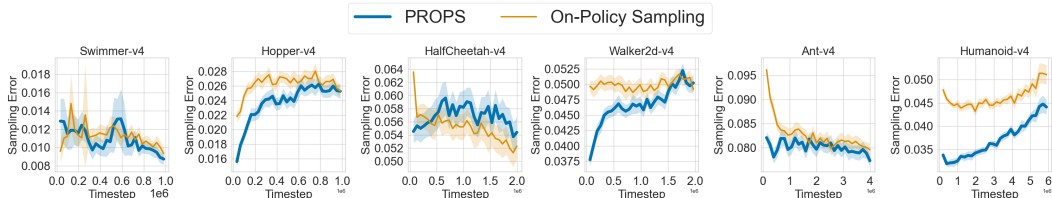

Figure 10: Sampling error throughout RL training. Solid curves denote the mean over 5 seeds. Shaded regions denote 95% confidence belts.

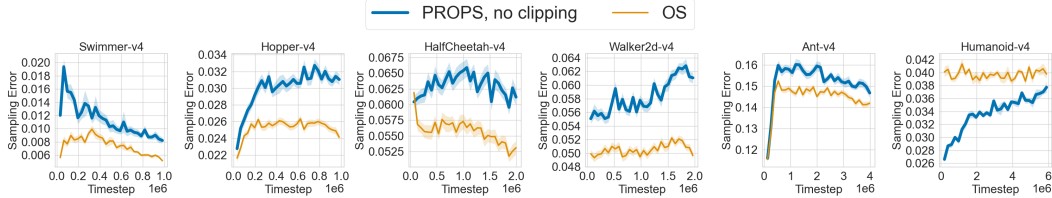

Figure 11: Sampling error throughout RL training without clipping the PROPS objective. Solid curves denote the mean over 5 seeds. Shaded regions denote 95% confidence belts.

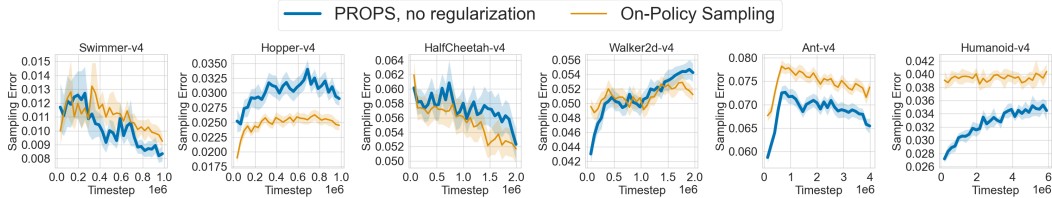

Figure 12: Sampling error throughout RL training without regularizing the PROPS objective. Solid curves denote the mean over 5 seeds. Shaded regions denote 95% confidence belts.

## D CORRECTING SAMPLING ERROR DURING RL TRAINING

In this appendix, we include additional experiments investigating the degree to which PROPS reduces sampling error during RL training, expanding upon results presented in Section 6.2 of the main paper. We include sampling error curves for all six MuJoCo benchmark tasks and additionally provide ablation studies investigating the effects of clipping and regularization on sampling error reduction and data efficiency in the RL setting.

As shown in Fig 10, PROPS achieves lower sampling error than on-policy sampling throughout training in 5 out of 6 tasks. We observe that PROPS increases sampling error but nevertheless improves data efficiency in HalfCheetah as shown in Fig. 5a. This result likely arises from our tuning procedure in which we selected hyperparameters yielding the largest return. Although lower sampling error intuitively correlates with increased data efficiency, it is nevertheless possible to achieve high return without reducing sampling error.

In our next set of experiments, we ablate the effects of PROPS's clipping mechanism and regularization on sampling error reduction and data efficiency. We ablate clipping by tuning RL agents with

$\epsilon_{\text{PROPS}} = \infty$, and we ablate regularization by tuning RL agents with $\lambda = 0$. Fig. 11 and Fig. 12 show sampling error curves without clipping and without regularization, respectively. Without clipping, PROPS achieves larger sampling than on-policy sampling in all tasks except Humanoid. Without regularization, PROPS achieves larger sampling error in 3 out of 6 tasks. These observations indicate that while clipping and regularization both help reduce sampling during RL training, clipping has a stronger effect on sampling error reduction. As shown in Fig. 13 PROPS data efficiency generally decreases when we remove clipping or regularization.

Lastly, we consider training with larger buffer sizes $b$ in Fig. 14. We find that data efficiency may decrease with a larger buffer size. Intuitively, the more historic data kept around, the more data that must be collected to impact the aggregate data distribution.

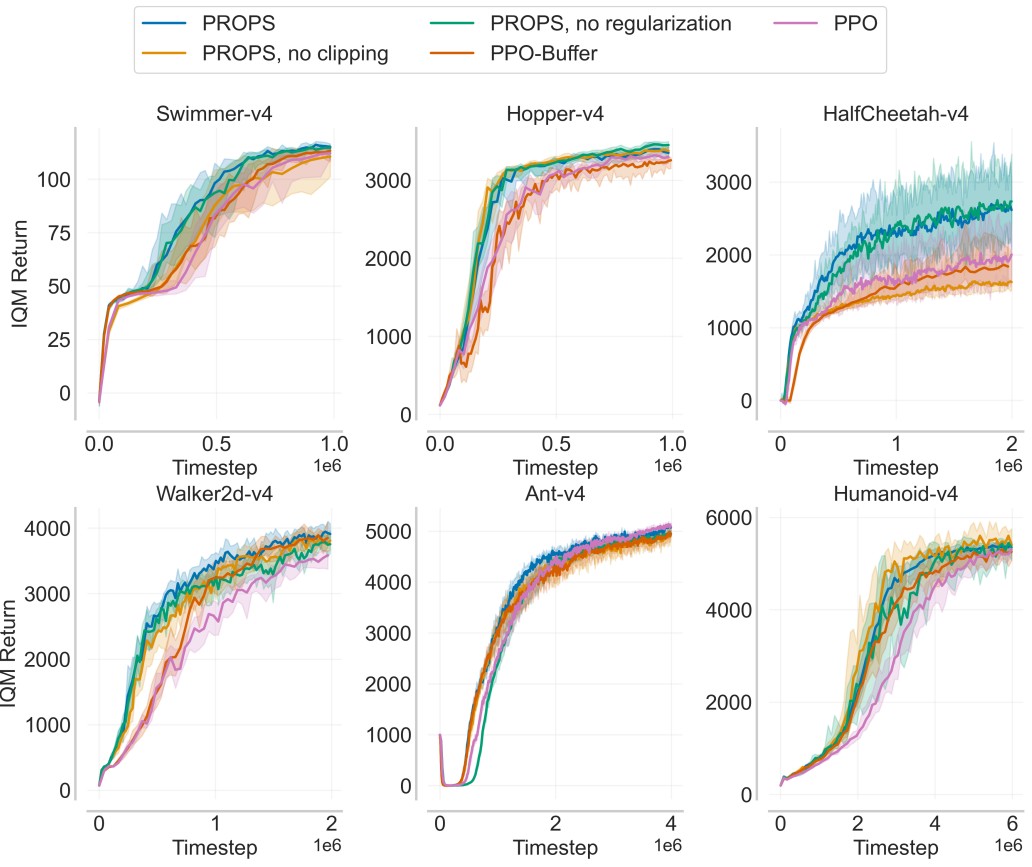

Figure 13: IQM return over 50 seeds of PROPS with and without clipping or regularizing the PROPS objective. Shaded regions denote 95% bootstrapped confidence intervals.

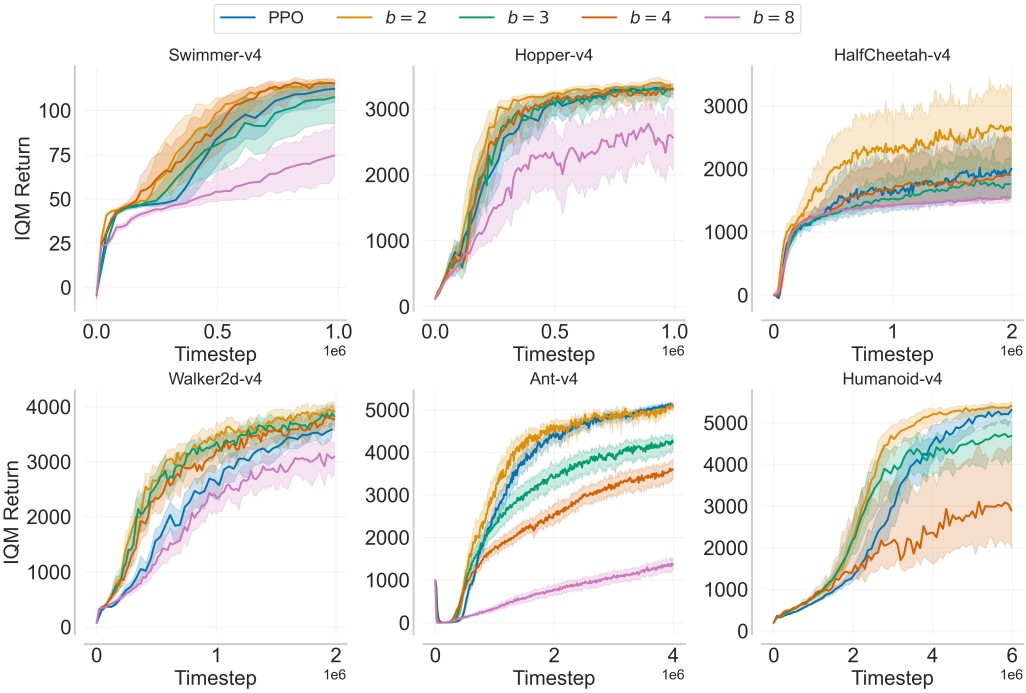

Figure 14: IQM return over 50 seeds for PROPS with different buffer sizes. We exclude $b = 8$ for Humanoid-v4 due to the expense of training and tuning. Shaded regions denote 95% bootstrapped confidence intervals.

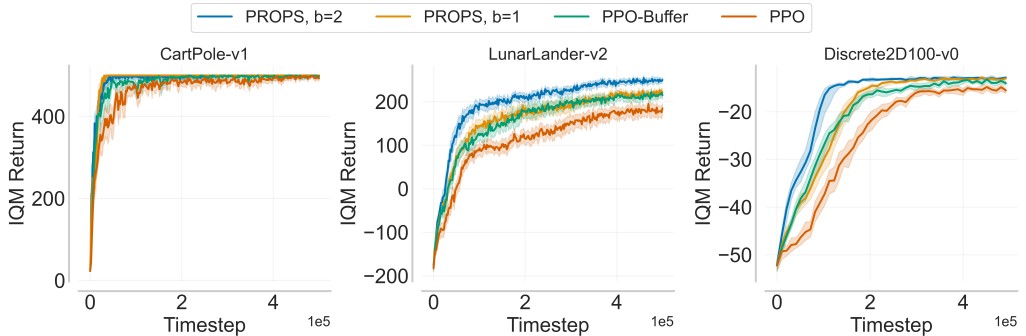

Figure 15: IQM return for discrete action tasks over 50 seeds. Shaded regions denote 95% boot-strapped confidence intervals.

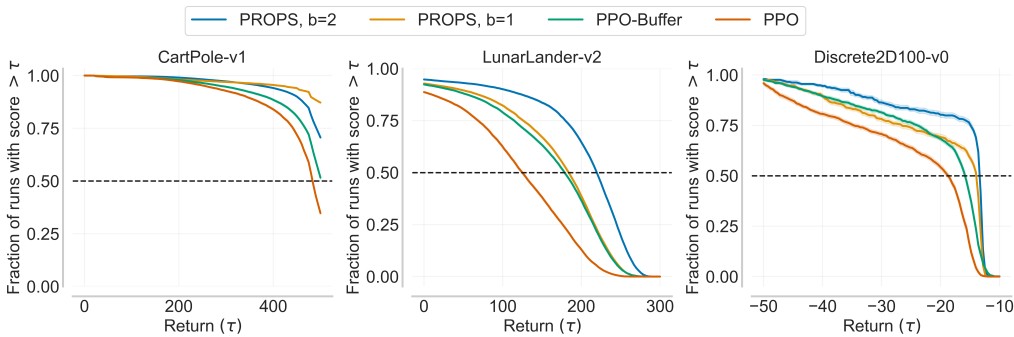

Figure 16: Performance profiles for discrete-action tasks over 50 seeds. The return $\tau$ for which the profiles intersect $y = 0.5$ is the median, and the area under the performance profile corresponds to the mean. Shaded regions denote 95% bootstrapped confidence intervals.

## E  DISCRETE-ACTION TASKS

We include 3 additional discrete-action domains of varying complexity. The first two are the widely used OpenAI gym domains CartPole-v1 and LunarLander-v2 (Brockman et al., 2016). The third is a 2D navigation task, Discrete2D100-v0, in which the agent must reach a randomly sampled goal. There are 100 actions, each action corresponding to different directions in which the agent can move. From Fig. 15 and 16 we observe that PROPS with $b = 2$ achieves larger returns than PPO and PPO-BUFFER all throughout training in all three tasks. PROPS with $b = 1$ (no historic data) achieves larger returns than PPO all throughout training in all three tasks and even outperforms PPO-BUFFER in CartPole-v1 and Discrete2D100-v0 even though PPO-BUFFER learns from twice as much data. Thus, PROPS can improve data efficiency *without* historic data.

| PPO learning rate | $10^{-3}, 10^{-4}$, linearly annealed to 0 over training |
| PPO batch size $n$ | $1024, 2048, 4096, 8192$ |
| PROPS learning rate | $10^{-3}, 10^{-4}$ (and $10^{-5}$ for Swimmer) |
| PROPS behavior batch size $m$ | $256, 512, 1024, 2048, 4096$ satisfying $m \leq n$ |
| PROPS KL cutoff $\delta_{\text{PROPS}}$ | $0.03, 0.05, 0.1$ |
| PROPS regularizer coefficient $\lambda$ | $0.01, 0.1, 0.3$ |

Table 2: Hyperparameters used in our hyperparameter sweep for RL training.

| | |
|---|---|
| PPO number of update epochs | 10 |
| PROPS number of update epochs | 16 |
| Replay buffer size $b$ | 2 target batches (also 3, 4, and 8 in Fig. 14) |
| PPO minibatch size for PPO update | $bn/16$ |
| PROPS minibatch size for ROS update | $bn/16$ |
| PPO and PROPS networks | Multi-layer perceptron |
| | with hidden layers (64,64) |
| PPO and PROPS optimizers | Adam (Kingma and Ba, 2015) |
| PPO discount factor $\gamma$ | 0.99 |
| PPO generalized advantage estimation (GAE) | 0.95 |
| PPO advantage normalization | Yes |
| PPO loss clip coefficient | 0.2 |
| PPO entropy coefficient | 0.01 |
| PPO value function coefficient | 0.5 |
| PPO and PROPS gradient clipping (max gradient norm) | 0.5 |
| PPO KL cut-off | 0.03 |
| Evaluation frequency | Every 10 target policy updates |
| Number of evaluation episodes | 20 |

Table 3: Hyperparameters fixed across all experiments. We use the PPO implementation provided by CleanRL (Huang et al., 2022).

## F HYPERPARAMETER TUNING FOR RL TRAINING

For all RL experiments in Section 6.2 and Appendix D), we tune PROPS, PPO-BUFFER, and PPO separately using a hyperparameter sweep over parameters listed in Table 2 and fix the hyperparameters in Table 3 across all experiments. Since we consider a wide range of hyperparameter values, we ran 10 independent training runs for each hyperparameter setting. We then performed 50 independent training runs for the hyperparameters settings yielding the largest returns at the end of RL training. We report results for these hyperparameters in the main paper. Fig. 17 shows training curves obtained from a subset of our hyperparameter sweep.

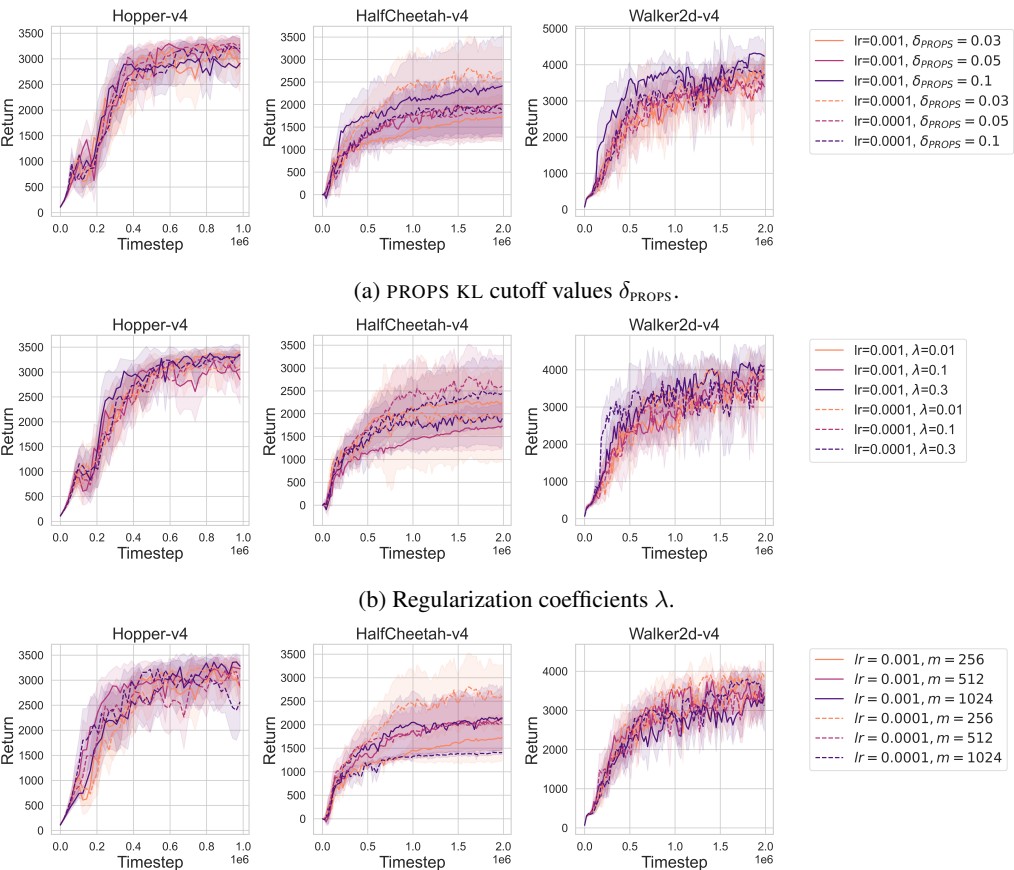

(a) PROPS KL cutoff values $\delta_{\text{PROPS}}$.

(b) Regularization coefficients $\lambda$.

(c) Behavior batch sizes $m$ (*i.e.* the number of steps between behavior policy updates).

Figure 17: A subset of results obtained from our hyperparameter sweep. Default hyperparameter values are as follows: PROPS KL cutoff $\delta_{\text{PROPS}} = 0.03$; regularization coefficient $\lambda = 0.1$; behavior batch size $m = 256$. Darker colors indicate larger hyperparameter values. Solid and dashed lines have the PROPS learning rate set to $1 \cdot 10^{-3}$ and $1 \cdot 10^{-4}$, respectively. Curves denote averages over 10 seeds, and shaded regions denote 95% confidence intervals.

Figure 18: Runtimes for PROPS, PPO-BUFFER, and PPO. We report means and standard errors over 3 independent runs.

## G  RUNTIME COMPARISONS

Figure 18 shows runtimes for PROPS, PPO-BUFFER, and PPO averaged over 3 runs. We trained all agents on a MacBook Air with an M1 CPU, use the same tuned hyperparameters used in throughout the paper. PROPS takes at most twice as long as PPO-BUFFER. These results agree with the intuition we shared in our initial response: Both PROPS and PPO-BUFFER learn from the same amount of data but PROPS learns two policies.

We note that PPO-BUFFER is faster than PPO is HalfCheetah-v4 because, with our tuned hyperparameters, PPO-BUFFER performs fewer target policy updates than PPO. In particular, PPO-BUFFER is updating its target policy every 4096 steps, whereas PPO is updating the target policy every 1024 steps.

## H  COMPARISON TO SAC

To help visualize the differences in data efficiency between PROPS and an off-policy algorithm, we compare against SAC, a state-of-the-art off-policy algorithm. We tune SAC by sweeping over the following batch sizes in $\{64, 128, 256\}$, actor learning rates in $\{1e - 3, 3e - 4\}$ and critic learning rates in $\{1e - 3, 3e - 4\}$. We report results for the best hyperparameters in Fig. 19. In all tasks except Swimmer-v4, SAC is more data efficient than PROPS. PROPS greatly outperforms SAC in Swimmer-v4. In Hopper-v4, PROPS is competitive with SAC and appears more stable.

To estimate the runtime of SAC agents, we record the time required to train SAC agents for 20k timesteps and then extrapolate out to the full training budget. Runtimes estimates for SAC as well as runtimes for PROPS, PPO-BUFFER, and PPO shown in Fig. 20 *in log scale*. Although SAC generally outperforms PROPS in terms of data efficiency, PROPS is subsantially faster in terms of wallclock time.

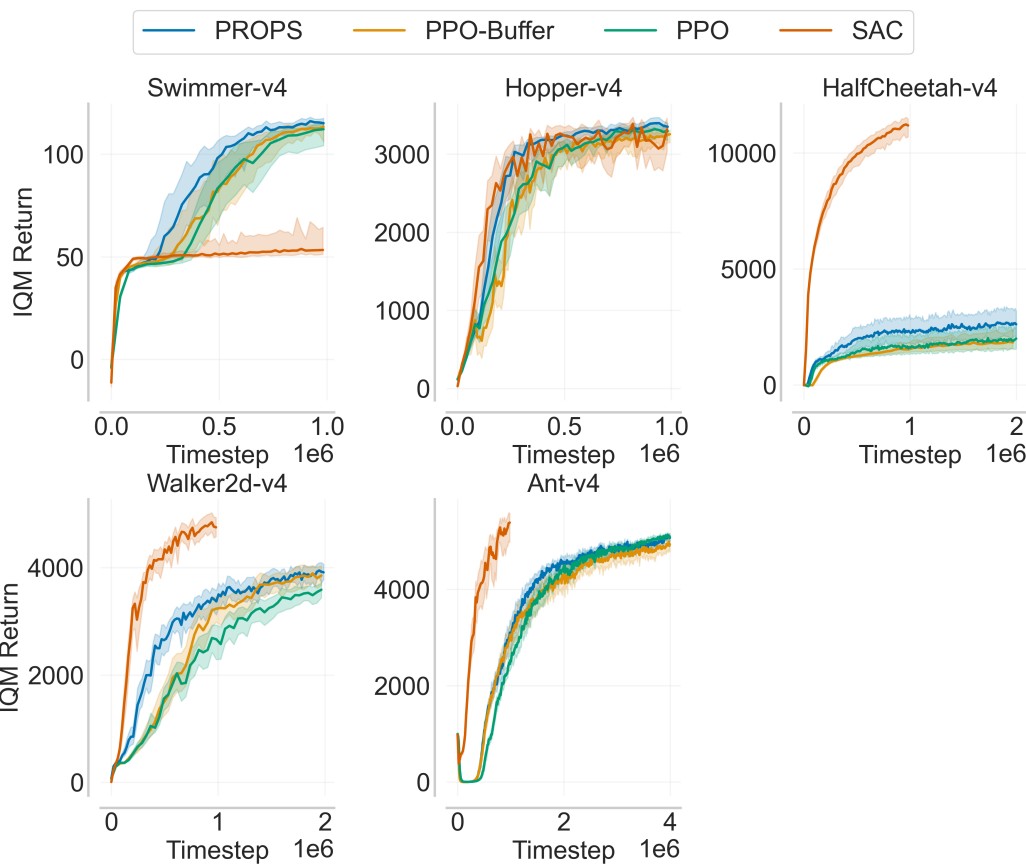

Figure 19: IQM returns over 50 seeds. Shaded regions denote 95% bootstrapped confidence intervals.

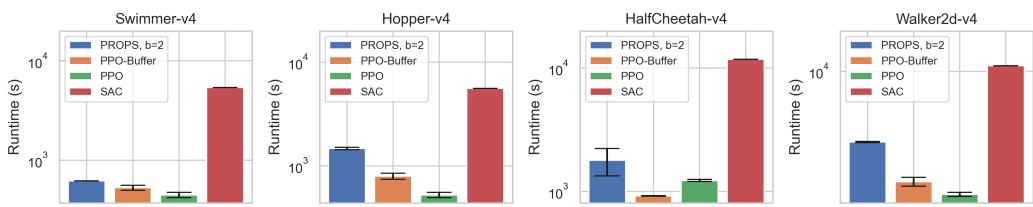

Figure 20: Runtimes for PROPS, PPO-BUFFER, PPO, and SAC. We report means and standard errors over 3 independent runs. Note that the vertical axis is log scale.

