# OpenReview forum: "On-Policy Policy Gradient Reinforcement Learning Without On-Policy Sampling"
_ICLR.cc/2024/Conference — Submitted to ICLR 2024_

### Official Review · Reviewer_KNrT · 2023-10-30

**Soundness:** 2 fair
**Presentation:** 2 fair
**Contribution:** 2 fair
**Rating:** 5
**Confidence:** 4

**Summary:**

The paper proposes a new algorithm, PROPS, which seeks to entail more efficient off-policy learning by drawing inspirations from on-policy learning algorithms, i.e., making the update data distribution stay close to the learner policy (on-policy). The paper provides a new algorithm, with experimental results that show improvements over baselines such as PPO.

**Strengths:**

The paper is interesting in that it tries to improve off-policy learning by drawing inspirations from on-policy learning algorithms, and the intuition that one should try to make the data to be as on-policy as possible. The paper hinges on an observation that for the learning to be on-policy, it suffices for the data distribution to be on-policy, and one does not always have to use the same policy for the update.

**Weaknesses:**

The idea of using near on-policy update or constrained update to improve off-policy learning has been extensively studied in the RL literature, and as a result, the constrained update rule proposed in the paper is not very novel.  Though the idea of using off-policy distribution that matches on-policy update is interesting, it does not appear enough to constitute a technically solid algorithmic contribution. Indeed, the main idea presented in the paper is not fundamentally different from the trust region policy updates, and does not present additional insight beyond what's already in the existing literature.

**Questions:**

=== **PPO objective and CLIP objective** ===

The PPO objective appears similar as the CLIP objective in Eqn 4. It is important to note that PPO takes $\theta$ and $\theta_{\text{old}}$ to be close by policy (in practice one iteration away from each other). In the limit when these two policies are very close, the update approximates the policy gradient update. In light of this, in order for the update to be sensible for CLIP, we also need $\theta$ and $\phi$ to be close to one another. In practice this would be softly enforced by the KL loss. Does that sound right? What would be the major difference between CLIP and PPO objectives, despite the technical differences on the clipping operation and KL regularizations.

=== **CLIP objective** ===

In Eqn 4 and 5 we specify a loss for both $\theta$ and $\phi$, do we update them both at the same time? Or is one updated at a slower rate compared to another.

If $\theta$ and $\phi$ are practically close to each other, then the CLIP objective is effectively very similar to the idea of trust region policy updates.

=== **On-policy and off-policy** ===

I like the comment that on-policy learning should really be to match the learning distribution, not the exact policy. However, it might not pay off much to read too much into the terminology itself -- I think much of the community would agree that the naive implementation of on-policy learning would be to learn from the same policy, whereas on-policy samples are all that's needed to perform the update. In practice since the policy keeps drifting away, it is very difficult to maintain even on-policy samples or sub-sample those from the replay buffer.

I think the paper has a very nice starting point to identify this conceptual insight, but didn't dive deep enough to address the issue. The CLIP objective is not fundamentally different from trust region policy updates, which have been in the literature since the beginning of deep RL.

=== **Comparison in Fig  5** ===

What's PPO-buffer exactly? Does it correspond to running PPO on off-policy samples naively? I'd feel that PROMPS is a slightly more adaptive variant of PPO-buffer, so that it can make use of buffer data as well. But I am surprised about the empirical gains, since PPO-buffer, though using off-policy data, still applies clipping and hence enforces a trust region constraint during the update. This means the policy update still maintains its stability over time, even when using very off-policy samples.

**Details Of Ethics Concerns:**

No concerns.

---

> ### Author Response · Authors · 2023-11-16
>
> We appreciate your thoughtful review and are glad that you found our approach to be interesting!
>
> We are slightly confused by some of these comments and hope we can clarify our novelty during the discussion period. We would like to emphasize that the core novelty of our work is how we demonstrate that *off-policy sampling* can reduce sampling error and improve the data efficiency of *on-policy* policy gradient algorithms. In other words, the core novelty is not only the PROPS update we use to adapt our behavior policy but moreover the fact that we are using a separate behavior policy for on-policy learning in the first place. So while KL penalties and clipping (as in PPO) are not novel on their own, our use of them to enable scalable adaptive sampling with the behavior policy is novel to the best of our knowledge.
>
> Below, we address your comments and questions.
>
> # **Comparing the PPO objective and the CLIP objective of PROPS**
>
> The major difference between these objectives is that CLIP is an objective for *behavior policy* optimization and the PPO objective is for optimizing the *target policy.* In CLIP, we weight behavior policy updates by -1 to increase the probability of under-sampled actions, where as PPO weights policy changes by the advantage $A(s, a)$ to increase the probability of good actions with large advantages. Note that we use the PPO objective to optimize the target policy in both PROPS and vanilla PPO.
>
> # **Are $\theta$ and $\phi$ close to each other?**
>
> Your understanding is correct. At the start of each PROPS update, we set $\phi \gets \theta$ (line 3 in Algorithm 2), and then make a location adjustment to $\phi$ that increases the probability of under-sampled actions. You are also correct in saying that regularization helps to keep the behavior policy close to the target policy and that the PROPS update is similar to a trust-region update.
>
> # **Are $\theta$ and $\phi$ updated at different frequencies?**
>
> The behavior policy is updated more frequently than the target policy so that the behavior policy has a chance to reduce sampling error before each target update. In our experiments, the behavior policy is updated every 256 timesteps and the target policy is updated every 1024 - 8192 timesteps.
>
> # **Why we emphasize the difference between on-policy data vs. on-policy sampling**
>
> While we agree that the formulation of the on-policy update does indeed point to on-policy data as the key desiderata, unfortunately, we know of many sources that distinguish on-policy learning by its sampling distribution, including some prominent educational resources:
>
> 1. Sutton & Barto’s RL textbook [1] states, “the distinguishing feature of on-policy methods is that they estimate the value of a policy while using it for control.”
> 2. Lecture 5 of David Silver’s RL course [2] states that on-policy algorithms “learn about policy π from experience sampled from π.”
> 3. The taxonomy of RL algorithms provided in OpenAI’s Spinning Up package [3] states that with on-policy algorithms, “each update only uses data collected while acting according to the most recent version of the policy.”
>
> Because the view that "on-policy refers to a sampling procedure" is so prominent, we wanted to elaborate on the distinction between on-policy data and on-policy sampling to emphasize how this popular perception is not a reality. We will clarify these points in our revisions and hope to further discuss them with reviewers.
>
> # **PPO-Buffer and its performance**
>
> Your understanding is correct; PPO-Buffer naively reuses historic off-policy data without any algorithmic changes, and PROPS is indeed an adaptive version of PPO-Buffer.
>
> PPO-Buffer outperforms PPO in Walker2d-v4 and Humanoid-v4, which indeed suggests that PPO’s clipping mechanism may be robust to stale data. In fact, OpenAI reused historic data in a similar fashion when using PPO to train the RL agents that ultimately defeated the world champions in the Dota 2 game [4]. However, we note that in most tasks we consider, PPO-Buffer performs comparably to PPO.
>
> Thank you again for your review, and please let us know if our response clarifies your comments and questions! We are happy to discuss any follow-up comments.
>
> # **References**
>
> [1] Sutton, Richard S., and Andrew G. Barto. "Reinforcement learning: An introduction." MIT press, 2018.
>
> [2] Silver, David. "RL Course by David Silver, Lecture 5: Model-Free Control." University College London, Computer Science Department, 2015. https://www.davidsilver.uk/wp-content/uploads/2020/03/control.pdf
>
> [3] Achiam, Joshua. Spinning Up in Deep Reinforcement Learning. Github, 2018. https://spinningup.openai.com/en/latest/spinningup/rl_intro2.html#a-taxonomy-of-rl-algorithms
>
> [4] Berner et. al. "Dota 2 with Large Scale Deep Reinforcement Learning." Arxiv 2019.

---

### Official Review · Reviewer_8MkC · 2023-10-31

**Soundness:** 2 fair
**Presentation:** 3 good
**Contribution:** 2 fair
**Rating:** 5
**Confidence:** 4

**Summary:**

This work focuses on sampling error problem of on-policy RL algorithms. Following the previous work ROS, this paper introduces the understanding: on-policy learning requires on-policy data, not on-policy sampling. This paper proposes a new method called Proximal Robust On-Policy Sampling (PROPS), aiming at adaptively correcting sampling error in previously collected data by increasing the probability of sampling actions that are under-sampled with respect to the current policy. Two more mechanisms are utilized to address the issues of destructive large policy updates and gaussian policy updates. The proposed method is evaluated in MuJoCo continuous control tasks and three discrete-action tasks, against PPO and PPO-Buffer, ROS.

**Strengths:**

- The paper is almost clear and overall well-written.
- Intuitive motivating examples are used which help smooth reading.
- The experiments are conducted with both learning performance comparison and sampling error analysis.

**Weaknesses:**

My main concern lies at the mismatch between *the idea* mentioned multiple times in the first half of the paper (i.e., adaptively corrects sampling error in previously collected data by increasing the probability of sampling actions that are under-sampled with respect to the current policy) and *the practical method* proposed later:

- The idea is to adapt the behavior policy so that the resultant data distribution can be more on-policy, as illustrated in Figure 1.
- However, the practical method is to adapt the behavior policy to fit the collected data (with the gradient $\nabla_{\phi}L=-\nabla_{\phi} \sum_{(s,a) \in D} \log \pi_{\phi}(a|s)$ and its improvement).

&nbsp;

For some specific point, the second paragraph of Section 5 says, ‘To ensure … updates to $\pi_{\phi}$ should attempt to increase the probability of actions which are currently under-sampled with respect to $\pi_{\theta}$… the gradient $\nabla_{\phi}L=-\nabla_{\phi} \sum_{(s,a) \in D} \log \pi_{\phi}(a|s)$ provides a direction to change $\phi$ such that under-sampled actions have their probabilities increased’. I do not get the point. Since the expectation is over the collected data, the gradient seems to be an MLE based on the collected data (or imitation learning) and it is almost the same as Equation 7 in the appendix (with a difference in max and min).

Please correct me if I misunderstand it.

&nbsp;

Besides, I feel that the example and discussion used in Section 4 are improper. The main idea of the paper and the illustrative example in Figure 1 focus on the distribution mismatch of *action*; while the discussion in Section 4 is mainly about the distribution mismatch of *outcome*, which cannot be known in practical learning process (i.e., we have no idea about the existence of a large reward). This turns to be a little bit off the track to me.

&nbsp;

The empirical results are not convincing enough to me, especially under the concerns I mentioned above.

Important baselines ROS and GePPO are not included in Figure 5. Computational cost like wall-clock time is not discussed in the main body of the paper. Besides, the ablation on the KL term in learning performance is expected.

**Questions:**

1) Since there could be under-sampled actions, there should be also over-sampled actions. How are they considered in the proposed method?

2) Why is ROS not included in learning performance comparison like Figure 5? In addition, I think it is necessary to include GePPO as a baseline, since the proposed method aims at improving learning efficiency of on-policy RL.

3) How is the computation cost of PROPS, especially the wall-clock time?

---

> ### Author Response · Authors · 2023-11-16
> **Author Rebuttal (1/2)**
>
> We would first like to thank the reviewer for their comments and questions; we’re glad you found our paper well-written and appreciated our motivating examples. We believe your comments can be easily addressed with small changes to our current paper. We describe these in our response below and kindly ask you to consider raising your rating.
>
> ## **Perceived mismatch between the idea and practical method**
>
> We believe this concern arises from a simple misunderstanding of the direction of the ROS/PROPS update, and we will clarify this point in the main paper. The key point is that we are doing gradient *ascent* on the negative log-likelihood (NLL) rather than gradient *descent* as done in MLE learning. You correctly point out that MLE learning would have a mismatch with the goal of our work. However, since we use gradient *ascent* on the NLL, the update has the desired effect of increasing the probability of under-sampled actions.
>
> It is important to note that the ROS and PROPS algorithms do not aim to fully optimize their objectives. Rather, at each iteration, the behavior policy is set equal to the target policy and then the gradient is used to make a local adjustment which decreases the probability of over-sampled actions, thereby increasing the probability of under-sampled actions.
>
> # **Example in Section 4**
>
> You are correct in that PROPS focuses on correcting the state-conditioned distribution of actions as opposed to the distribution of trajectories (since we generally don’t know the trajectory distribution). To convey intuition for how sampling error could impact learning, our example in Section 4 focused on sampling error in the outcomes space. However, we see how this was confusing to multiple reviewers and so we can make a few minor modifications to make the example focus on action probabilities.
>
> In particular, we can add the following action probabilities to Figure 2:
>
> $\pi(left | s_0) = 0.5$,
>
> $\pi(right | s_0) = 0.5$,
>
> $\pi(left | s_L) = 0.02$,
>
> $\pi(right | s_L) = 0.98$
>
> Where, for brevity, we use $left$ = “attempt to move” and $right$ = “remain standing”. These action probabilities preserve the outcome probabilities originally listed.
>
> After 100 visits to $s_0$, the agent may never sample the $left$ branch at $s_L$ and believe that the $left$ branch at $s_0$ always leads to a return of -1 (falling). Having observed a return of +1 (standing) in the $right$ branch at $s_0$, the agent reduces the probability of choosing the $left$ branch at $s_0$, ultimately making the agent more likely to converge to a suboptimal standing policy. We can help correct this sampling error by increasing the probability of sampling the under-sampled $left$ branch at $s_L$.
>
> We hope our response clarifies the soundness of the ROS/PROPS updates as well as the example in section 4. If you have further comments or questions regarding our empirical results, we would be more than happy to discuss further!
>
> # **Baselines**
>
> We considered these baselines but ultimately decided to exclude them from the main RL experiments shown in Figure 5.
>
> We exclude ROS from our RL experiments for two reasons:
> 1. Since ROS fails to reduce sampling error even when the target policy is fixed (Fig. 3 in the main paper and Fig. 7 and 8 in Appendix C of the supplemental material), we expect ROS to also fail to reduce sampling error in the more difficult RL setting where the target policy is continually updated.
> 2. ROS is too expensive to run in the RL setting. ROS updates the behavior policy *every timestep* using a gradient computed over *all data* in the agent’s buffer, whereas PROPS updates the target policy every 256 timesteps using minibatches sample from the agent’s buffer.
>
> We can clarify both of these reasons in our revisions.
>
> PROPS and GePPO are orthogonal approaches to improve the data efficiency of on-policy learning: PROPS uses adaptive data collection to reduce sampling error, while GePPO reweights data according to historic policies *after* data collection. We ultimately decided that the comparison distracted from our main focus of determining whether adaptive off-policy sampling could produce sufficiently on-policy data for effective learning. For the purpose of answering the main questions of our study, the empirical evaluation is complete as-is.

---

> ### Author Response · Authors · 2023-11-16
> **Author Rebuttal (2/2)**
>
> # **Ablations on KL regularizer and clipping mechanism**
>
> We ablate PROPS’s KL regularizer and the clipping mechanism in Appendix C and D and find that both help reduce sampling error and improve agent performance:
> * Appendix C, Fig. 9 shows how clipping and regularization affect PROPS’s ability to reduce sampling error with respect to a *fixed* target policy. In this setting, clipping generally has a stronger effect on sampling error reduction than regularization.
>
> * Appendix D, Fig 12 shows how clipping and regularization affect PROPS’s ability to reduce sampling error in the RL setting with a *continually changing* target policy. In this setting, both regularization and clipping are crucial to reducing sampling error.
>
> * Appendix D, Fig 13 shows IQM return achieved by PROPS during RL training with and without regularization or clipping. In nearly all tasks, PROPS performs best with both clipping and regularization.
>
> # **Wall-clock time comparisons**
>
> We can include wall-clock time comparisons of PROPS, PPO, and PPO-Buffer in our revisions. We will share our results in a follow-up comment once we have them.
>
> Intuitively, we expect PROPS will take roughly twice as much time as PPO-Buffer; both algorithms perform updates using the same amount of data, though PROPS learns two policies.
>
> Again, please let us know if our response clarifies your comments and questions! We will happily discuss any follow-up comments you may have.

---

> > ### Comment · Reviewer_8MkC · 2023-11-21
> > **Response to Authors' Rebuttal**
> >
> > I appreciate the response provided by the authors. Some of my concerns are addressed.
> >
> > For one more questions, the author mentioned 'Rather, at each iteration, the behavior policy is set equal to the target policy and then the gradient is used to make a local adjustment which decreases the probability of over-sampled actions, thereby increasing the probability of under-sampled actions' in the response. According to Algorithm 1, it seems that the behavior policy is set to the target policy only once at the beginning. Is it a mistake? Is line 4 ought to be between line 6 and line 7?

---

> > > ### Author Response · Authors · 2023-11-21
> > >
> > > We appreciate the discussion! Below, we respond to your question below, and we also include the runtime comparison requested in your initial review.
> > >
> > > We believe there is a minor confusion regarding Algorithm 1. As you've noted, we initialize the behavior policy equal to the target policy at the beginning of training (Line 4 in Algorithm 1). However, we also set the behavior policy equal to the target policy at the start of each PROPS update (**Line 3 in Algorithm 2**). The purpose of Line 4 in Algorithm 1 is to initialize the behavior policy for first batch of data collection, whereas Line 3 in Algorithm 2 is a core feature of the PROPS update. We can modify Line 10 in Algorithm 1 to say "Update $\pi_\phi$ with $\mathcal D$ using Algorithm 2" to emphasize this distinction.
> > >
> > > Please let us know if our response clarifies this point!
> > >
> > > # Runtime comparison
> > >
> > > The following table shows runtimes for PROPS, PPO-Buffer, and PPO averaged over 3 runs with standard deviations in parentheses. We have results for 4 environments, and we can add the two remaining environments (Ant-v4 and Humanoid-v4)  in a camera-ready version. We trained all agents on a MacBook Air with an M1 CPU, and we use the same tuned hyperparameters used in throughout the paper.
> > >
> > > | Environment | PROPS | PPO-Buffer | PPO |
> > > | -------------- | -----------  | ----------- | ------ |
> > > | Swimmer-v4 | 618 (1)  | 529 (30) | 446 (30) |
> > > | Hopper-v4   | 1473 (40) | 795 (54) | 525 (30) |
> > > | HalfCheetah-v4 | 1786 (44) | 921 (6) | 1228 (23) |
> > > | Walker2d-v4 | 2963 (33) | 1500 (112) | 1203 (42) |
> > >
> > > PROPS takes at most twice as long as PPO-Buffer. These results agree with the intuition we shared in our initial response: Both PROPS and PPO-Buffer learn from the same amount of data but PROPS learns two policies.
> > >
> > > We note that PPO-Buffer is faster than PPO is HalfCheetah-v4 because, with our tuned hyperparameters, PPO-Buffer performs fewer target policy updates than PPO. In particular, PPO-Buffer is updating its target policy every 4096 steps, whereas PPO is updating the target policy every 1024 steps.
> > >
> > > Again, thank you for the discussion! If there are any lingering questions or concerns, please let us know!

---

### Official Review · Reviewer_ErDF · 2023-10-31

**Soundness:** 3 good
**Presentation:** 3 good
**Contribution:** 4 excellent
**Rating:** 8
**Confidence:** 4

**Summary:**

This paper proposes a new adaptive sampling approach to reducing sampling error in off-policy policy optimization. Specifically, the paper proposes to collect data such that the data distribution is more on-policy to the target policy. They do this by training a separate behavior policy according to a objective function which encourages the behavior to take actions which correct the sampling distribution of the replay buffer. They show some empirical improvements in standard benchmarks compared to PPO.

**Strengths:**

Overall, I think the proposed method is quite reasonable and an interesting approach to correcting for the use of off-policy data in policy optimization. I see two weaknesses in the paper as currently presented.

**Weaknesses:**

I am updating my score based on conversations w/ the authors. Specifically, I appreciate the re-focus and the additions for clarity.


--- before edits ----

Overall, I think the proposed method is quite reasonable and an interesting approach to correcting for the use of off-policy data in policy optimization. I see two weaknesses in the paper as currently presented.

## Characterization of on-policy updates (i.e. data or sampling)

I believe the paper presents an inaccurate characterization of the history of on-policy vs off-policy algorithms, and this detracts from the overall presentation of the paper. The core of my issue is the characterization that the field is uncertain if on-policy updates require on-policy data or on-policy sampling.

To understand my complaint, I want to begin with policy evaluation and think about how these principle ideas transitioned into generalized policy iteration and concepts surrounding the usual suspects of on-policy learning (i.e. those using “on-policy sampling”). If we look at the fundamentals of policy evaluation in dynamic programming we see a very clear picture of the ideal update (i.e. using the probabilities of the state distribution and policies directly), but at the cost of knowing the transition probabilities and having to sweep over all states and actions for a single update. When improving the policy we go between a policy improvement step (i.e. maximize the learned value function) and the policy evaluation step. Generalized policy iteration made this update more general by enabling the ability to improve a policy without running policy evaluation for every state and action.

From generalized policy iteration we can get to many of our on-policy algorithms such as TD, sarsa, on-policy actor-critic algorithms, and many others with the inclusion of sampling from distributions instead of knowing the distributions ahead of time. This type of on-policy learning uses a transition and throws it away, meaning any policy improvement done will always appear in the data _and_ there is no stale buffer of data. From this trajectory of literature it is clear, on-policy algorithms are designed and work well with data that is distributed according to the target policy. The easiest way to get this data is through sampling according to the behavior.

With the inclusion of replay buffers, even on-policy algorithms are likely to have off-policy drifting in the data used to train them. We see these problems accumulate in bad behavior (see Deadly Triad) and hacks (e.g. Target Network) designed to navigate this problem to try and get more out of the data an agent experiences.

The paper correctly notice a difference in the sampling error when the data isn’t infinitely large, where the adaptive sampling method presented here is perfectly suited to fill this gap and try and correct for the discrepancies in the distribution of actions (which might influence the distribution of states, but I’m unclear how this influence would evolve).

To wrap this up, I don’t have a complaint about the method, and am personally very excited about more active techniques of the agent modifying its behavior to get more favorable data distributions. I just think the narrative is misleading, and a dichotomy presented by the authors is ill-conceived and distracting.

## Example in section 4
2. The example used in section 4 is quite confusing in the context of off-policy and on-policy policy optimization. The main confusion from my perspective is the confusion between a trajectory and an action, which propagates throughout the paper in that we are only correcting the distribution according to the action distribution and no the state distribution.


## Some suggestions and questions about the empirical section:
1. How were the hyperparameters chosen for all the methods?
2. I think the empirical section would benefit from the inclusion of an off-policy baseline to see how much progress is being made by the adaptive sampling method. While you don’t necessarily have to beat this baseline, including SAC could be beneficial to interpreting the results of your method!

**Questions:**

See above.

---

> ### Author Response · Authors · 2023-11-15
>
> Thank you for the in-depth comment on the framing of our work. We completely agree with the historical framing you provide,  so we believe a minor change can clarify the narrative. However, before doing so, we would like to explain why we framed the paper as we did and discuss how we can combine your suggestions with our original motivation.
>
> While we agree that the historical trajectory does indeed point to on-policy data as the key desiderata for on-policy policy gradient methods, unfortunately, we know of many cases where on-policy learning is defined by the sampling distribution, including some prominent educational resources:
> 1. Sutton & Barto’s RL textbook [1] states, “the distinguishing feature of on-policy methods is that they estimate the value of a policy while using it for control.”
> 2. Lecture 5 of David Silver’s RL course [2] states that on-policy algorithms “learn about policy π from experience sampled from π.”
> 3. The taxonomy of RL algorithms provided in OpenAI’s Spinning Up package [3] states that with on-policy algorithms, “each update only uses data collected while acting according to the most recent version of the policy.”
>
> Because the view that "on-policy refers to a sampling procedure" is so prominent, we chose a more pointed narrative to emphasize that it is the empirical distribution of data that really matters for accurate gradient estimates. In a sense, our work is pointing back to the historical trajectory you give and saying that we really do want to have samples weighted according to their actual probability and not just generated according to a specific process.
>
> Overall, we believe that there is not a large gap between our view and the reviewer’s view and that we can fine-tune the discussion in our paper to note the historical trajectory given. Specifically, it will be straightforward to 1) add discussion on the historical basis of these methods and say that, yes, clearly on-policy data is the goal, and 2) clarify that we aim to correct a popular misconception in how these algorithms are understood. We hope that we can discuss this point more with the reviewer over the next week to make the narrative as strong as possible.
>
> [1] Sutton, Richard S., and Andrew G. Barto. Reinforcement learning: An introduction. MIT press, 2018.
>
> [2] Silver, David. "RL Course by David Silver, Lecture 5: Model-Free Control." University College London, Computer Science Department, 2015. https://www.davidsilver.uk/wp-content/uploads/2020/03/control.pdf
>
> [3] Achiam, Joshua. Spinning Up in Deep Reinforcement Learning. Github, 2018. https://spinningup.openai.com/en/latest/spinningup/rl_intro2.html#a-taxonomy-of-rl-algorithms

---

> > ### Comment · Reviewer_ErDF · 2023-11-15
> >
> > Great finds! Yes after further consideration, you are correct in that there seems to be a misleading nature to some of these conversations when moving to the case of acting in an environment which I have forgotten (and/or misplaced in my head). I love the adaptive sampling approach, so am more than happy to come to some middle ground here to make the paper's narrative better.
> >
> > I think the crux of this comes from the fact that when jumping to the RL case (i.e. no longer in dynamic programming land). When jumping to sampling (of any kind) there will always be differences in the empirical and true distributions, but when we discuss acting in the environment in RL this isn't communicated or addressed in the current curriculum most students are exposed to. I believe the best way to address that in this paper would be to discuss the historical context briefly (i.e. how we go from DP to sampling approaches) and discuss how the empirical distribution and true distributions are likely not the same within the confines of the agent's finite life (i.e. our infinite data assumptions are bogus and we should deal with it).
> >
> > I totally agree with the author on all their points, and now better understand where they were coming from. I think the pointed approach could work here, but you need to show mastery on the historical context of your statements to prevent people like me from getting grumpy :). While I think this can work, I think getting to the heart of your motivation "the empirical and true distributions are not the same in a finite life" would be more straightforward and would not come off as denigrating prior work (just my 2 cents).
> >
> > I think this could thread the needle quite well and likely won't need a considerable amount of work. If this can be fixed, the comments from multiple authors about the example in section 4 addressed, and some of the comments about the clarity in the experimental section, then this paper would be a great contribution (even for a future conference if it doesn't get in here).

---

> > > ### Author Response · Authors · 2023-11-22
> > >
> > > Thank you again for the thoughtful comments! We've uploaded a revised submission that we believe address your comments about the narrative. We decided to take your suggestion and focus on the heart of our motivation for PROPS:
> > > 1. We've reframed the introduction to focus on finite sampling error in on-policy learning.
> > > 2. Instead of asking whether on-policy algorithms require on-policy sampling or on-policy data, we now *state* that on-policy algorithms require on-policy data and that on-policy sampling is simply one way to acquire it.
> > >
> > > We're glad to see that you think our work would make a great contribution :) Please let us know if this any concerns remain about the narrative!
> > >
> > > We've also included SAC as a baseline in Appendix H. In general, SAC is more data efficient than PROPS but 5-10x slower in terms of wall clock time. In Hopper-v4, PROPS is fairly competitive with SAC. Humanoid-v4 results are still pending; SAC agents take much longer to train on this task with our computing resources. We can include Humanoid results in a camera-ready version of this work.

---

> > > > ### Comment · Reviewer_ErDF · 2023-11-22
> > > > **Response**
> > > >
> > > > Thank you for your effort! I believe this makes the paper much better, and prevents onerous people like me from getting grumpy :)! I will update my score accordingly.

---

> > > > > ### Author Response · Authors · 2023-11-22
> > > > >
> > > > > We're glad to see the updated score! Thank you once more for your comments and discussion; they were quite helpful for crafting a cleaner narrative.

---

> ### Author Response · Authors · 2023-11-16
> **Author Rebuttal (2)**
>
> We greatly appreciate your response regarding our clarification of the motivating narrative! We'll provide another update on that soon. In the meantime, we wanted to use this separate post to respond to your other comments regarding the example in section 4 and a few experimental details.
>
> # **Example in section 4**
>
> You are correct in that PROPS focuses on correcting the state-conditioned distribution of actions as opposed to the distribution of trajectories (since we generally don’t know the trajectory distribution). To convey intuition for how sampling error could impact learning, our example in Section 4 focused on sampling error in the outcomes space. However, we see how this was confusing to multiple reviewers and so we can make a few minor modifications to make the example focus on action probabilities.
>
> In particular, we can add the following action probabilities to Figure 2:
>
> $\pi(left | s_0) = 0.5$,
>
> $\pi(right | s_0) = 0.5$,
>
> $\pi(left | s_L) = 0.02$,
>
> $\pi(right | s_L) = 0.98$
>
> Where, for brevity, we use $left$ = “attempt to move” and $right$ = “remain standing”. These action probabilities preserve the outcome probabilities originally listed.
>
> After 100 visits to $s_0$, the agent may never sample the $left$ branch at $s_L$ and believe that the $left$ branch at $s_0$ always leads to a return of -1 (falling). Having observed a return of +1 (standing) in the $right$ branch at $s_0$, the agent reduces the probability of choosing the $left$ branch at $s_0$, ultimately making the agent more likely to converge to a suboptimal standing policy. We can help correct this sampling error by increasing the probability of sampling the under-sampled $left$ branch at $s_L$.
>
> Please let us know if further clarification is needed on this updated example!
>
> # **Hyperparameters:**
>
> We tune all algorithms using a hyperparameter sweep described in Appendix F (Tables 2 and 3). We perform this sweep separately for all tasks and all algorithms and report results for top-performing hyperparameters.
>
> # **Off-policy baseline:**
>
> We agree that it would be interesting to compare PROPS to an off-policy algorithm. We’ll have to get back to you on this matter; we first want to integrate your feedback on the narrative we crafted.

---

### Official Review · Reviewer_FJW4 · 2023-11-01

**Soundness:** 4 excellent
**Presentation:** 3 good
**Contribution:** 3 good
**Rating:** 8
**Confidence:** 4

**Summary:**

This paper looks at the problem of using replay buffers with on-policy methods. The replay buffer contains data generated from various (past) policies but on-policy methods (in practice) rely on data being on-policy.

Building on [Zhong et al., 2022](https://proceedings.neurips.cc/paper_files/paper/2022/file/f2dbede0879b9d04ceb30f1b8b476b27-Paper-Conference.pdf), the paper explores adapting the behavior policy (the policy generating the data) in such a way that the replay, as a whole, looks on-policy with respect to the target policy (the policy being learned). (In this sense, the problem being studied can be seen as using on-policy methods for off-policy learning.)

While Zhong et al., 2022, considered the problem of policy evaluation, this paper considers the control problem. The proposed method, PROPS, works as follows:
train a policy to generate "diverse" data, with data from the replay. This policy solves an RL problem where the reward is -1 for taking the observed action and zero otherwise.
generate data with the policy from step 1.
train the target policy with data from the replay.

The policy in (1) is trained not to be too different from the target policy (the policy being trained in 3), but the the regime in (1) is slightly off-policy. The regime in (3) is also slightly off-policy. So the paper proposes to use PPO for both steps (1) and (3) as a way to mitigate the slight off-policiness.

The paper shows that the data generated by PROPS is more on-policy than the data generated by PPO. Moreover, PROPS outperforms PPO and the "ad-hoc" PPO with a replay in a control benchmark. Importantly, PROPS works well with the replay whereas the ad-hoc PPO with a replay works worse than PPO in some of the tasks and better in others.

Overall I am happy with this paper, but I am divided in my assessment of its significance. On the one hand, it is a good study of the problem of making on-policy methods work with replays (and thus become more sample efficient). On the other hand, it is unclear to me whether we should use PROPS, or other adapted on-policy methods instead of an off-policy method when learning with a replay (e.g., MPO).

**Strengths:**

The paper is well executed, and the method is described in enough detail that I feel confident I could reproduce it.

**Weaknesses:**

I don't think the paper has any serious flaws, but there are some areas it can be improved:
* the presentation can be improved to emphasize some points about the paper and the main contributions
* the significance of the results should be discussed in more detail

I note in passing that the paper is not specifically concerned with representation learning.

**Questions:**

### Comments

Thank you for your submission.

I think the introduction can be clarified a bit more for the distinction between on-policy sampling and on-policy data. My guess is that the focus on the distinction is not as helpful as getting to the point that on-policy methods need on-policy data for the updates, and the data collection should take into account what the data distribution in the replay will look like.  In the paper you show that if the method samples data on-policy the replay ends up looking quite off-policy.

I would even consider using PPO-buffer as the example to motivate the issue. "Consider the following method. Repeat: Add data to a replay with latest policy, update the latest policy with PPO updates. This method clearly samples the data on-policy, because the behavior and target policy are the same before the policy updates are applied. However, the data in the replay may not be on-policy, because the replay buffer collects data generated by various past policies. Therefore the example method may fail at policy improvement because PPO updates rely on near on-policy data in order to work as intended." A diagram of a replay buffer with "chunks" of data from various policies might also help quickly understand the issue.

Along these lines, I think the point in the beginning of section 4 would be easier to get across if you used an example with a replay buffer where the data comes from multiple policies. I think trying to argue that a sample from a single policy may look off-policy is trickier, and in practice it's not so much of a problem (PPO works just fine with on-policy sampling to generate the data for each update).

I partially disagree with paragraph 5 of section 4. I don't think off-policy corrections should be dismissed as "may increase variance". Some off-policy corrections that clip importance weights mitigate gradient variance in exchange for some bias, for example, PPO. I would say that the real problem is that, while off-policy corrections can mitigate off-policiness, methods that use them typically still suffer as data gets more off-policy. So they may still benefit from different sampling strategies for collecting experience, even with off-policy corrections in place.

I think there are typos in Eq. 4. I think the minus sign should be outside of clip.

In my opinion the KL penalty in Eq. 5 and the KL rule in line 9 of Alg. 2 are under-investigated. In my experience the KL regularizer in Eq. 5 contributes to make collected data more on-policy, so I think it would be helpful to understand how much the KLs are contributing to the replay buffer being on-policy.

We can also see the impact of the KL in Figure 3. The KL regularization actually increases the on-policiness of the data. In contrast, even though ROS was meant (by motivation) to also increase the on-policiness of the data, it does not. It is also worth noting that making the data more on-policy might also mean that the clipping and regularization are not letting the updated target policy deviate too much from the pre-update target policy. So maybe with regularization even on-policy sampling might generate more on-policy data.

Another point on ROS: I find the idea of the ROS loss to generate more diverse data a bit delicate to work with, in the sense that you want a policy that generates a bit less of the actions seen, but not completely optimizing the -1 reward. If you end up rewarding the agent with -1 all the time, the learning dynamics might take you to a uniform policy, but for the problem itself any policy is an optimal policy.

I suggest placing Figure 5 earlier, close to where it is discussed.

Please consider sorting the references alphabetically instead of by citation order, as it will be easier to refer to.

### Questions

What is the contribution of the paper in a broader context? The contributions are clear to me for the problem of on-policy methods with a replay, but what about the broader problem of increasing sample-efficiency by using a replay? Do you think adjusted on-policy methods with a replay are a suitable alternative to off-policy methods?

---

> ### Author Response · Authors · 2023-11-18
>
> We would like to thank the reviewer for their positive review! We’re pleased to see you appreciate our contributions towards improving the data efficiency of on-policy learning.
>
> We want to begin by answering your question about the work’s broader impact. The core contribution of our work is that we demonstrate how *off-policy sampling* can reduce sampling error and improve the data efficiency of *on-policy* policy gradient algorithms. Our work also emphasizes that on-policy algorithms simply require on-policy data and are agnostic to how the data is generated – a point which is often overlooked in the literature.
>
> Below, we now address your comments and questions.
>
> # **The distinction between on-policy sampling and on-policy data in the Introduction**
>
> We framed the introduction in this way to emphasize that the focus with on-policy algorithms should be on the data, not the procedure for generating the data. In the literature, we find that this point is often overlooked. For instance,  Sutton & Barto’s RL textbook [1] states that “the distinguishing feature of on-policy methods is that they estimate the value of a policy while using it for control.”
>
> # **ROS update**
>
> This is a great point that we will clarify in the main paper. ROS is indeed delicate to work with, and a key aspect of our our is work is designing a method (PROPS) which makes ROS more stable in higher dimension benchmark tasks. PROPS, like ROS, does not aim to fully optimize its objective but instead uses the gradient to make a local adjustment to the behavior policy that increases the probability of under-sampled actions. PROPS’s clipping mechanism and KL regularization help ensure that this behavior update remains local.
>
> # **Ablations on KL regularizer and clipping mechanism**
>
> We ablate PROPS’s KL regularizer and the clipping mechanism in Appendix C and D and find that both help reduce sampling error and improve agent performance:
>
> * Appendix C, Fig. 9 shows how clipping and regularization affect PROPS’s ability to reduce sampling error with respect to a *fixed* target policy. In this setting, clipping generally has a stronger effect on sampling error reduction than regularization.
>
> * Appendix D, Fig 12 shows how clipping and regularization affect PROPS’s ability to reduce sampling error in the RL setting with a *continually changing* target policy. In this setting, both regularization and clipping are crucial to reducing sampling error.
>
> * Appendix D, Fig 13 shows IQM return achieved by PROPS during RL training with and without regularization or clipping. In nearly all tasks, PROPS performs best with both clipping and regularization.
>
> # **Should we prefer PROPS over other off-policy methods?**
>
> We view on-policy learning with replay as a means to improve the data efficiency of on-policy learning rather than an alternative to off-policy learning. On-policy and off-policy algorithms have different advantages and disadvantages, and the choice of algorithm typically depends on what advantages are most relevant for a given task. For instance, PPO has played a role in several RL success stories [*e.g.*, 2, 3] in part because it is empirically stable and robust to hyperparameter choices.
>
> # **Example in Section 4**
>
> While we agree that an example with a replay buffer would clearly illustrate how historic off-policy data can bias updates, we wanted to emphasize that sampling error can be a problem even when all data in our buffer comes from the current target policy. In fact, in Fig. 15 and 16 of Appendix E in the supplemental material, we show that PROPS can improve the data efficiency of on-policy learning even when the replay buffer contains *no historic data* (PROPS with b=1).
>
> # **Suggested edits and typos**
>
> Thanks for pointing out all of these! We’ll fix these in our revision:
>
> * We will clarify the limitation of off-policy approaches under large distribution shift.
> * We will try to keep figures on the same page on which they are referenced.
> * You’re correct that minus should be outside the clip function in PROPS’s clipped objective--nice catch
> * We will order references alphabetically.
>
> Please let us know if you have any follow-up questions or comments. We are more than happy to discuss!
>
> # **References**
>
> [1] Sutton, Richard S., and Andrew G. Barto. "Reinforcement learning: An introduction." MIT press, 2018.
>
> [2] Berner et. al. "Dota 2 with large scale deep reinforcement learning." Arxiv 2019.
>
> [3] Vinyals et. al. "Grandmaster level in starcraft ii using multi-agent reinforcement learning." Nature 2019.

---

### Author Response · Authors · 2023-11-20
**Global Response**

We would like to thank all reviewers for their helpful feedback and comments! We are glad that reviewers found our proposed method to be interesting (ErDF, KnRT), our empirical analysis to be sound (8Mkc, ErDF), and our paper well-written (FJW4, 8MkC).

A few reviewers (ErDF, KnrT, FJW4) suggested revising the introduction to focus less on “on-policy sampling vs on-policy data” distinction, since it is already clear that on-policy algorithms require on-policy data and are agnostic to how the data is generated. We framed the introduction in this way to emphasize that this point is often overlooked in the literature. For instance, Sutton & Barto’s RL textbook [1] states that “the distinguishing feature of on-policy methods is that they estimate the value of a policy while using it for control.” After some discussion with reviewer ErDF, we believe that we can clarify this point with minor revisions (and reviewer ErDF seems to agree). Please see the discussion below. We do hope to discuss these points with reviewers KNrT and FJW4 as well.

Reviewers ErDF and KNrT also pointed out that our method corrects sampling error in the distribution of *actions* whereas our motivating example in Section 4 focuses on sampling error in the distribution of *outcomes*. We thought that focusing on outcomes would make it easier to understand how sampling error could impact learning, though we now see how this could be confusing. In our responses, we described how we can make this example focus on action distributions without fundamentally changing the example by simply adding action probabilities to Fig. 2.

The remaining reviewer comments were primarily clarification questions about our algorithm, and we believe we have addressed these. We'd like to invite the reviewers to discuss further comments or questions they may have; we will happily discuss them with you!

[1] Sutton, Richard S., and Andrew G. Barto. "Reinforcement learning: An introduction." MIT press, 2018.

---

### Author Response · Authors · 2023-11-22
**Global Response 2: Revisions**

Dear reviewers,

Thank you again for the helpful feedback! We've uploaded a new submission with a few minor revisions based on reviewer suggestions and comments. Revisions are shown in blue text. The core revisions are as follows:

* **Section 1:** We've reframed the introduction to focus less on the "on-policy sampling vs on-policy data" distinction and more on how finite on-policy sampling inevitably results in sampling error. To do this, we swapped the order of paragraphs 1 and 2 and made a few revisions to the (new) second paragraph.
* **Section 4:** The motivating example now focuses on sampling error in the distribution of actions rather than outcomes. The example is fundamentally the same; we simply added action probabilities to Fig. 2 and replace some references to trajectories with references to actions.
* **Section 6.2** We've clarified why we omit ROS from our RL experiments: ROS is expensive and fails to reduce sampling error even with a fixed policy.

We hope to engage in discussion with all reviewers; please let us know if our responses have clarified your comments! If there are any lingering questions, we're happy to discuss!

---

### Comment · Area_Chair_BT9U · 2023-11-22
**A comment on the way to update behavioral policy.**

Regarding reviewer 8MkC's concern, that is, there is a mismatch between the intended idea and the actually taken approach. I think the authors may adopt the following explanation. This is only the AC's personal thought, and thus is not mandatory.

Suppose data distribution at this point is $\lambda(s,a) = \mu(s)\times\nu(a|s)$. Then the (weighted) KL divergence between $\nu(\cdot|s)$ and $\pi_b(\cdot|s)$ will be

$|D|\sum_{s}\mu(s)KL(\nu(\cdot|s),\pi_\phi(\cdot|s)) = |D|\sum_{s}\mu(s)\sum_{a}\nu(a|s)\log(\nu(a|s)/\pi_\phi(a|s)) = \sum_{(s,a)\in D}-log \pi_\phi(a|s) + const$

where "const" depends only $\mu$ and $\nu$, but is irrelevant to $\phi$.

Therefore, updating in the direction of $-\sum_{(s,a)\in D}\nabla log \pi_\phi(a|s)$ accounts for one-step of gradient ASCENT to INCREASE the weighted KL divergence:  $|D|\sum_{s}\mu(s)KL(\nu(\cdot|s),\pi_\phi(\cdot|s))$. In general, to increase KL divergence, one would like to increase $\pi_\phi(a|s)$ if $\pi_\phi(a|s)\geq\nu(a|s)$ and decrease it if the "<" happens. This means that if the data buffer appears to observe action $a$ taken at $s$ fewer than what $\pi_b(a|s)$ indicates, then increase $\pi_b(a|s)$.

The authors may also think of a more detailed illustrative m-armed bandit as a single state MDP. Consider tabular softmax behavioral policy $\pi_\phi(a|s) = exp(\phi_{a})/\sum_{i}exp(\phi_{i})$. Then one can exactly compute the gradient update as
$$\phi_a \leftarrow \phi_a + \alpha(-\nu(a) + \pi_\phi(a))$$
which says increase $\phi_a$ if $a$ is under sampled and decrease $\phi_a$ if $a$ is over sampled.

---

> ### Author Response · Authors · 2023-11-22
>
> We’d like to thank the AC for this explanation! We considered a similar formulation when developing our methods, and we can try to include some discussion on this in a camera-ready version.

---

### Meta-Review · Area_Chair_BT9U · 2023-12-04

**Metareview:**

This paper proposes an empirical approach to adaptively adjust the sampling policy in order to "correct" the mismatch between the data buffer and the policy of interest. The basic idea of "on-policy RL methods need on-policy data instead of on-policy sampling" is interesting and worth discussion, which is a good attempt to making on-policy methods work with replays buffers. However, as the reviewers pointed out, the paper also lacks solid theoretic validation. For example, it is not clear whether the currently proposed approach will converge or not. As the current paper with constantly changing "target policy" cannot be covered by the theory in (Zhong et. al., 2018), which focused on evaluating a fixed policy.

SAC note: the more outstanding problem to me is the disconnection between the story and the proposed method. The paper's story is that one should use replay buffer to "mimic" on-policy PG --- which people have done exactly (see the off-policy PG literature, e.g., https://arxiv.org/pdf/1904.08473.pdf) by (marginalized) importance sampling. However, if I skip all the way to method (Eq.4 and 5), we see a variant of PPO with behavior regularization, which one can propose completely without any connection to the "on-policy PG from replay" discussion and motivation.

Another point is that there is a subtle distinction between PG and PI. PI can be easily run off-policy, but has stronger assumption on the structure of the policy class (namely, every time the improvement policy greedy wrt the previous Q need to be represented by the policy parameterization). PG needs to be run on-policy (at least in its standard form), but the benefit is that it is more agnostic to policy parameterization --- it will just find a local optimum in the policy class. I am raising this distinction because since the final algorithm looks more like approximate PI than PG, and PI just doesn't care that much about on-policyness, so again one can explain the method & empirical success in a way that is largely orthogonal to the "on-policy PG" story.

**Justification For Why Not Higher Score:**

The paper lacks enough theoretical discussion of the current result.

**Justification For Why Not Lower Score:**

The paper proposes an interesting idea that "on-policy RL methods need on-policy data instead of on-policy sampling".

---

### Decision · Program_Chairs · 2024-01-16

Reject